



# Impact of mesoscale eddies on water mass and oxygen distribution in the eastern tropical South Pacific

Rena Czeschel[1], Florian Schütte[1], Robert A. Weller[2], Lothar Stramma[1]

[1]GEOMAR Helmholtz Centre for Ocean Research Kiel, Düsternbrooker Weg 20, 24105 Kiel, Germany
[2]Woods Hole Oceanographic Institution (WHOI), 266 Woods Hole Rd, Woods Hole, MA 02543, USA

*Correspondence to*: R. Czeschel (rczeschel@geomar.de)

**Abstract.** The influence of mesoscale eddies on the flow field and the water masses, especially the

oxygen distribution of the eastern tropical South Pacific is investigated from a mooring, float and satellite data set. Two anticyclonic (ACE1/2), one mode water (MWE) and one cyclonic eddy (CE) are identified and followed in detail with satellite data on their westward transition with velocities of 3.2 to 6.0 cm/s from their generation region, the shelf of the Peruvian and Chilean upwelling regime, across the Stratus Ocean Reference Station (ORS) (~20°S, 85°W) to their decaying region far west in the

oligotrophic open ocean. The ORS is located in the transition zone between the oxygen minimum zone and the well-oxygenated South Pacific subtropical gyre. Velocity, hydrographic, and oxygen measurements at the mooring show the impact of eddies on the weak flow region of the eastern tropical South Pacific. Strong anomalies are related to the passage of eddies and are not associated to a seasonal signal in the open ocean. The mass transport of the four observed eddies across 85°W is between 1.1

and 1.8 Sv. The eddy type dependent available heat, salt and oxygen anomalies are $7.6 \times 10^{18}$J (ACE), $0.8 \times 10^{18}$J (MWE), $-9.4 \times 10^{18}$J (CE) for heat, $23.9 \times 10^{10}$kg (ACE2), $-3.6 \times 10^{10}$kg (MWE), $-42.8 \times 10^{10}$kg (CE) for salt and $-3.6 \times 10^{16}$ μmol (ACE2), $-3.5 \times 10^{16}$ μmol (MWE), $-6.5 \times 10^{16}$μmol (CE) for oxygen showing an imbalance between anticyclones and cyclones for heat and salt transports probably due to seasonal variability of water mass properties in the formation region of the eddies. Heat, salt and

oxygen fluxes out of the coastal region across the ORS region in the oligotrophic open South Pacific are estimated based on these eddy anomalies and on eddy statistics (gained out of 23 years of satellite data).



Furthermore, four profiling floats were trapped in the ACE2 during its westward propagation between the formation region and the open ocean, which allows conclusions on the isolation of water mass properties and the lateral mixing with time between the core of the eddy and the surrounding water showing the strongest lateral mixing between the seasonal thermocline and the eddy core during the first

half of the lifetime.

## 1 Introduction

The eastern tropical South Pacific (ETSP) containing the Peruvian upwelling regime, which is one of the four major eastern boundary upwelling systems, shows pronounced meso- and submesoscale variability (e.g. Capet et al., 2008; McWilliams et al., 2009; Chaigneau et al., 2011). Mesoscale

variability in the ocean occurs as linear Rossby waves and as nonlinear vortices or eddies. During the last two decades eddies have been recognized to play an important role in the vertical and horizontal transport of momentum, heat, mass and chemical constituents of seawater (e.g., Chelton et al., 2007; Klein and Lapeyre 2009) and therefore contribute to the large-scale water mass distribution. Especially in upwelling areas, eddies have been identified as major agents for the exchange between coastal waters

and the open ocean (e.g. Chaigneau et al., 2008; Pegliasco et al.; 2015, Schütte et al., 2016a). At least three types of eddies have been identified: cyclonic, anticyclonic and anticyclonic mode water eddies (e.g., McWilliams, 1985; D'Asaro, 1988; McGillicuddy Jr. et al., 2007) as well as a transition from cyclonic eddies to 'cyclonic thinnies' to exist throughout the world ocean (McGillicuddy Jr. 2015). Usually, isopycnals in anticyclonic eddies are depressed for the entire vertical extent of the eddy while

in mode-water eddies a thick lens of water deepens the main thermocline while shoaling the seasonal thermocline (McGillicuddy Jr. et al., 2007). Mode water eddies are also often referred as intrathermocline eddies (ITEs) (Hormazabal et al., 2013). Cyclones dome both the seasonal and main pycnocline.

Several analyses of the mean eddy properties offshore the Peruvian coast are conducted in the last

decade and found the largest eddy frequency in the ETSP off Chimbote (~9°S) and south of San Juan (15°S) (e.g. Chaigneau et al., 2008 or Fig. 1a). Using a combination of Argo float profiles and satellite data the three-dimensional mean eddy structure of the eastern South Pacific was described for the

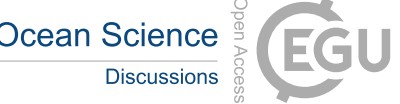


temperature, salinity, density, and geostrophic velocity field of cyclones as well as anticyclones (Chaigneau et al., 2011). However, a distinction in 'regular' anticyclones and anticyclonic mode water eddies is still pending in the ETSP. From recent findings (Stramma et al., 2013, Schütte et al., 2016a, Schütte et al., 2016b) it seems to be mandatory to distinguish between these two eddy types as they

strongly differ in their efficiency to transport conservative tracers, especially in upwelling areas. In addition it is observed that the different eddy types influence non-conservative tracers, like dissolved oxygen, on different ways within their isolated eddy cores. Especially cyclones and anticyclonic mode-water eddies have been reported to create an isolated biosphere, which greatly differs from the biosphere present in the surrounding areas (Altabet et al., 2012; Löscher et al., 2015). In these eddy

cores the oxygen concentration can decrease with time (Fiedler et al., 2016; Schütte et al., 2016b). In wide areas of the world oceans eddies with an open ocean low-oxygen core are observed (e.g. North Pacific: Lukas and Santiano-Mandujano, 2001; South Pacific: Stramma et al., 2013; tropical North Atlantic: Karstensen et al., 2015). These low-oxygen eddies have strong impacts on sensible metazoan communities and marine life (Hauss et al., 2016). Anammox is the leading nitrogen loss process in

ETSP eddies whereas denitrification was undetectable (Callbeck et al., 2017), while denitrification appears only patchy in the ETSP (Dalsgaard et al., 2012). Low-oxygen eddies release a strong negative oxygen anomaly during their decay, which may influence the large-scale oxygen distribution (Schütte et al., 2016b). A description of the oxygen structure for the ETSP dependent on the eddy-type (cyclone, anticyclone and mode-water eddy) is to our knowledge not reported so far.

The following paper includes an analysis of the three different eddy types and their impact on the water masses and oxygen distribution in the ETSP and is based on the Stratus Ocean Reference Station (ORS) mooring. The Stratus mooring is located at ~20°S, 85°W in the transition zone between the oxygen minimum zone (OMZ) and the well-oxygenated subtropical gyre (e.g. Tsuchiya and Talley, 1998). Eddy analyses were also done in the past at the Stratus mooring, where a snapshot of a strong

anticyclonic mode water eddy was observed in March/April 2012 (Stramma et al., 2014). In this paper we investigate in more detail the mooring period March 2014 to April 2015 and set a focus on the isolation and development of an eddy core during its isolated lifetime. We intensively followed eddies of each kind (two 'regular' anticyclones, one anticyclonic mode water eddy and one cyclone) which



crossed the Stratus mooring position, from their formation areas near the coast to their decay eastwards of the Stratus mooring (Fig. 1). During their lifetime one of these eddies were also partly sampled by several profiling floats equipped with oxygen sensors which were deployed in March 2014 within the eddies (Fig. 1).

In general, the large-scale oxygen distribution in the ETSP is dominated by a strong OMZ at depths of 100-900 m (e.g., Karstensen et al., 2008; Paulmier and Ruiz-Pino, 2009 or Fig. 1b). In the ETSP the zonal tropical current bands supply oxygen ($O_2$) rich water to the OMZ (Stramma et al., 2010). In contrast, the mid-depth circulation in the eastern South Pacific Ocean is sluggish in the region of the OMZ. As the mean currents are weak, eddy variability strongly influences the flow and ultimately

supplies oxygen-poor water to the OMZ (Czeschel et al., 2011). A rough estimate of the oxygen budget of the eastern tropical Pacific ocean (Stramma et al., 2010) was used to estimate 22% by vertical mixing, 33% by advection and the largest component of 45% by eddy mixing (Brandt et al., 2015).

The mean upper ocean circulation of the ETSP is relatively complex exhibiting several surface and subsurface currents. It is described to be composed out of the South Pacific subtropical gyre with the

north-eastern current band shown to be located south of 10 to 15°S called Humboldt Current, South Equatorial Current, Oceanic Chile-Peru Current or Peru Oceanic Current (e.g. Kessler 2006; Ayon et al., 2008), a set of several zonal current bands between the subtropical gyre and the equator (.e.g. Kessler 2006; Czeschel et al., 2015) as well as poleward and equatorward current bands near the South American continent (e.g. Chaigneau et al., 2013). The shipboard zonal velocity component along about

86°W composed of three ADCP surveys showed larger regions with westward then eastward flow between 13°S and 22°S (Brandt et al., 2015), however with influence by eddy features in ADCP measurements in November 2012 (Czeschel et al., 2015).

In general most of the eddies in the ETSP propagate westward originating from eddy generation hotspots near the coast following different eddy corridors (Fig. 1a). Coastal water properties are

captured within the eddy-cores and transported on their way into the open ocean across several oxygen, temperature, salinity and gradients (Fig. 1 b, c, d). The coastal water mass properties differ, due to the upwelling, which is strongest in the austral winter months from a seasonal cycle. The upwelled water



near the coast identified as Equatorial Subsurface Water (ESSW) (e.g. Thomsen et al., 2016) is colder, fresher and less oxygenated in austral winter than in austral summer.

This paper describes the temperature, salinity and oxygen anomaly of the different eddy types in the ETSP and their efficiency to dissipate the existing gradients. Of special interest is the eddy type

dependent isolation of the eddy cores during different eddy life stages. Knowledge about the initial eddy-core conditions near the generation areas, measurements during the mid-age of the eddy due to Argo floats and measurements of the Stratus mooring at the end of the eddy lifetime allows us to investigate the lateral mixing from the eddy-core water masses with its surrounding waters.

## 2 Data sets

### 2.1 Stratus mooring

Since October 2000 the Stratus mooring has been maintained at about 20°S, 85.5°W mainly to collect an accurate record of surface meteorology and air-sea fluxes of heat, freshwater, and momentum (Colbo and Weller, 2009). In addition velocity, pressure, temperature, conductivity sensors (for salinity computation) and 13 oxygen sensors were added to the mooring within the water column during the

deployment period 8 March 2014 to 25 April 2015 at 19°37'S, 84°57'W (velocity sensors used in this paper were also added during the deployment period 6 April 2011 to 29 May 2012 at 19°41'S, 85°34'W). The depth distribution of the different measuring devices for the 2014 to 2015 deployment period are given in Supplement Table S1 (depth distribution of the velocity sensors for the 2011 to 2012 deployment period are given in Stramma et al. (2014)). Annual mean velocity profiles were computed

for the upper 600 m for the two Stratus mooring deployment periods 2011/2012 and 2014/2015 where oxygen measurements were conducted. To avoid influence of a seasonal signal only the period 10 April to 9 April of the following year was computed and only instruments used which recorded the velocity for the entire period. These mean velocity profiles can be compared with the October 2000 to December 2004 mean velocity components (Colbo and Weller, 2007).

From the 13 oxygen sensors added to the 2014/2015 mooring period (Supplement Table S1), three instruments recorded erroneous oxygen values, which could not be corrected after the recovery. The





remaining ten oxygen sensors consist out of eight Aanderaa oxygen sensors in SeaGuard instruments, which were used with the manufacturers calibration (accuracy <8 µmol kg$^{-1}$ or 5%) and two oxygen-loggers, which received an additional lab-calibration. For the 15 MicroCats (pressure, temperature and salinity), a data calibration is done against shipboard CTD data during the service cruises (RV Ron

Brown RB 14-01 and RV Cabo de Hornos) and later by comparison with the data overlap with the previous mooring and by returning the instruments to SeaBird for laboratory calibration. The SeaGuard conductivity sensors in 107 m and 350 m depth have an offset of -0.13 psu and -0.18 psu, respectively.

## 2.2 Satellite data

Aviso (Archiving, Validation, and Interpretation of Satellite Oceanographic) satellite derived sea level

anomalies (SLA) data were obtained and used to identify and track the different eddies passing the Stratus mooring and to document the position of the floats within the eddies. The Copernicus Marine and Environment Monitoring Service (CMEMS, http://marine.copernicus.eu) has taken over the whole processing and distribution of the products formerly distributed by AVISO with no changes in the scientific content. The delayed-time "all-sat-merged" reference dataset of SLA is used which is mapped

on an ¼° x ¼° Cartesian grid and has a temporal resolution of one day. The time period January 1993 to December 2015 were chosen for the SLA and the geostrophic velocity anomalies also provided by AVISO.

For sea surface temperature (SST) the "Microwave Infrared Fusion Sea Surface Temperature" from Remote Sensing System (www.remss.com) is used. The data consist out of SST measurements of all

operational microwave (MW) radiometer (TMI, AMSR-E, AMSR2, WindSat) and infrared (IR) SST measurements (Terra MODIS, Aqua MODIS). Considered are here daily data with 9 km resolution from January 2013 to December 2015.

For sea surface chlorophyll (Chl) the MODIS/Aqua Level 3 data product mapped on a 4 km grid available at http://oceancolor.gsfc.nasa.gov provided by the NASA is used. The time period January

2013 to December 2015 with a daily resolution is chosen. Note, that the Chl data is cloud dependent as it is measured via IR.





### 2.3 Argo floats

Seven profiling Argo floats with Aanderaa oxygen sensors were deployed in March 2014 at 19°36'S, 84°58'W; 19°27'S, 83°01'W; 19°15'S, 80°30'W and 18°58'S, 76°59'W. The deployment locations (Fig. 1a) were chosen to be close to anticyclonic or cyclonic eddies determined from SLA figures. The

floats were deployed in pairs with drifting depth at 400 and 1000 dbar and cycling intervals of 10 days, except for 18°58'S, 76°59'W were only one float was deployed at 400 dbar drifting depth. From that seven Argo floats, four floats remained for a longer period within eddies which later crossed the Stratus mooring and are therefore used in more detail for our calculations in the paper (the four Argo floats are: 6900527, 6900529, 6900530 and 6900532). Typically a full calibration of the oxygen sensors on the

Argo floats is not available. The different manufactures of Argo float oxygen sensors specify their measurement error at least better than 8 µmol kg$^{-1}$ or 5%. Additionally the Argo float profiles of temperature, salinity and oxygen are compared and calibrated against the measurements of the Stratus mooring and against each other giving a relative accuracy.

### 3 Methods

From the Stratus mooring time series of velocity, temperature, salinity and oxygen (from 8 March 2014 to 25 April 2015) eddies of each type are identified and followed back and forward in time with the help of satellite data. The focus is set on one mode water eddy (MWE), two anticyclonic eddies (ACE1, ACE2) and one cyclonic eddy (CE) as they are also sampled by Argo floats (including oxygen sensors) in before.

#### 3.1 Heat, salt, and oxygen anomaly at the Stratus mooring

Available heat, salt and oxygen anomalies (AHA, ASA and AOA) were calculated as described in Chaigneau et al. (2011) and Stramma et al. (2014). At the Stratus mooring eddy core anomalies were estimated by the difference between the mean of temperature, salinity and oxygen within the eddy

boundaries and the background field estimated from the annual mean for the period 10 April 2014 to 9 April 2015. Eddy boundaries are determined for every depth by the mean of the maximum absolute





values of the maximum 90 h low-pass filtered southward and northward velocity. The mean westward propagation of the eddies estimated from SLA measurements is used to convert the time axis to a space axis leading to a mean radius. The vertical extent is defined as the depth of the coherent structure of the eddy, which is the ratio between the swirl velocity U and the propagation velocity c of the eddy. If U/c

> 1, the feature is nonlinear and maintains its coherent structure while propagating westward (Chelton et al., 2011). The swirl velocity is derived from the mean of the absolute values of the maximum 90 h low-pass filtered southward and northward velocity.

At the time when the mooring was deployed, part of the MWE had already passed the mooring. Therefore, the measurements of the eddy were mirrored to obtain the full coverage of the MWE.

**3.2 Determining of properties of the MWE, ACE1/2 and CE conducted from satellite data**

The eddy shape is identified by analysing streamlines of the SLA-derived geostrophic flow around an eddy centre (high/low SLA). An eddy boundary is defined as the streamline with the strongest swirl velocity (for more information on such an eddy detection algorithm see e.g. Nencioli et al., 2010). Note that the identified areas are irregularly circular therefore the circle-equivalent area is used to estimate

the eddy radius. Due to the resolution of the SLA data the eddy radius must be at least 45 km to unambiguously state that the identified area is a coherent mesoscale eddy and not an artificial signal. Clearly identified individual eddies may have a smaller radius than 45 km to get tracked. Eddies are tracked forward and backward in time following the approach described by Schütte et al. (2016a). To estimate the percentage of eddy coverage in the ETSP eddies are identified and tracked between 1993

and 2015. In the following it was counted how often a grid point (0.5° x 0.5°) was covered by an eddy structure. For the identification of eddy generation areas every newly detected eddy closer than 600 km off the coast is counted in 1° x 1° boxes. The sum of all these boxes is taken to compute the seasonal cycle of eddy generation. The Argo float profiles and the mooring time series are separated into data conducted within cyclones, anticyclones and the "surrounding area" which is not associated with eddy-

like structures also following the approach of Schütte et al. (2016a). In addition the relative position of the mooring or Argo float profile in relation to the eddy center and eddy boundary could be computed.





Furthermore, the composites of the eddy surface signatures (SLA, SST and Chl) consist of 150 x 150 km snapshots around the identified eddy centres. To exclude large-scale variations, the used SST data is low-pass filtered (cut-off wavelength of 15° longitude and 5° latitude) and subtracted from the original data to preserve only the mesoscale variability (see Schütte et al., 2016a for more details).

## 4 Results

### 4.1 General eddy generation and its seasonal cycle in the ETSP

In the ETSP 5244 eddies (49% cyclones; 51% anticyclones) are found between January 1993 to December 2015 (requirement: having a radius between 45 km and 150 km and visible for more than 7 days). Both types of eddies have an average radius of about 70 km and on average 15 % of the ETSP

are covered everyday with eddies (Fig. 1a). Most of the eddies are generated close to the Peruvian or Chilean coast, where large horizontal/vertical shears exist in an otherwise quiescent region. In almost entirely consistence with Chaigneau et al. (2008), hotspot locations of eddy generation are near the coast around 10°S and between 16°S to 22°S (Fig. 2a, b). The four eddies (MWE, CE, ACE1, and ACE1) described in detail below originate from the latter region. After their generation near the coast

the anticyclonic eddies tend to propagate north-westward whereas cyclonic vortices migrate south-westward (e.g. Chaigneau et al., 2008) into the open ocean. The seasonal cycle of eddy generation, based on all eddy new detections closer than 600 km off the coast, peaks in March and has its minimum in September (Fig. 2c), whereas cyclonic eddies exhibit a stronger amplitude. However, both anticyclonic as well as cyclonic eddies have their seasonal peak of formation in austral summer/fall

(February/March) and the lowest number at the end of austral spring (September; Fig. 2d).

The full eddy generation mechanisms are complex, whereby boundary current separation due to a sharp topographic bend is one important aspect of the eddy formation (Molemaker et al., 2015; Thomsen et al., 2016). It is suggested that anticyclones are generated due to instabilities of the Peru Chile Undercurrent (PCUC), whereas cyclonic eddies are formed from instabilities of the equatorward surface

currents (Chaigneau et al., 2013). In this context the strength of the PCUC is essential (Thomsen et al., 2016). Observations as well as models show a weak seasonal variability of the PCUC off Peru which is stronger in austral summer and fall (Thomsen et al., 2016; Chaigneau et al., 2013; Penven et al., 2005)





and might explain the higher number of eddy generation during this season. Other model simulations have revealed a seasonal cycle in eddy flux that peaks in austral winter at the northern boundary of the OMZ, while it peaks a season later at the southern boundary (Vergara et al., 2016). The PCUC also experiences relatively strong fluctuations with periods of a few days to a few weeks (Huyer et al.,

5   1991).

Intraseasonal and interannual variability is another factor modulating the strength of the PCUC off Chile and therefore the formation rate of anticyclonic eddies (Shaffer et al., 1999). El Niño (La Niña) events deepen (shoal) the thermocline and intensify (weaken) the PCUC (e.g. Montes et al., 2011; Combes et al., 2015). Although the strength of the PCUC increases during El Niño events the transport

of mode-water eddies decreases, which is due to a weakened baroclinic instability related to the deepening of the thermocline (Combes et al., 2015).

### 4.2 Eddy observations from March 2014 to April 2015 at Stratus mooring

From March 2014 to April 2015 the Stratus mooring was located at 84°57'W, 19°37'S, about 1500 km

offshore in the oligotrophic open ocean. Oxygen, salinity and meridional velocity component time series for the upper 600 m (Fig. 3; Supplement Fig. S1) record the passage of several eddies between March 2014 and April 2015. These observations are in agreement with the satellite data (SLA, SST and Chl) at the mooring location and the 450 to 295 m geopotential anomaly (Fig. 3a).

At the time of the mooring deployment on 8 March 2014 an anticyclonic mode water eddy (MWE) with

a radius of 43 km passed westward with the eddy centre to the north of the mooring while a cyclonic eddy was located south of the mooring site (Supplement Movie M1). The mooring instruments recorded the parameter distribution at the southern rim of the MWE and the typical oxygen, salinity and density distribution for mode water eddies: low oxygen of 13.4 µmol and high salinity (temperature) of 34.59 psu (9.47°C) in the eddy core in 350 m depth as well as upward bending of isopycnals above and

downward bending beneath the eddy core (Fig. 3b, c). In late March 2014 the MWE had moved westward.





A strong oxygen decrease as well as a salinity increase with a strong downward displacement of the isopycnals at mid-depth in early August 2014 was related to an anticyclonic eddy (eddy ACE1; Fig. 3b) that passed the mooring south of it. In the upper 250 m the oxygen concentration increased with the maximum about 10 days later than the oxygen minimum at 290 to 600 m depth (Supplement Fig. S1).

At 183 m depth, maxima in oxygen, temperature and salinity of 265 µmol/kg, 17.73°C and 35.38 psu were reached in mid-August 2014 reflecting the deepening of the pycnocline which brings warmer, more saline and oxygen-rich waters to deeper levels. The ACE1 shows meridional velocities of more than 5 cm/s in the upper 300 m depth (Fig. 3d).

Another strong oxygen decrease influenced the oxygen distribution from early November 2014 to early

January 2015. This anticyclonic eddy (eddy ACE2) had a radius of 53 km and showed the strongest downward displacement of the isopycnals in the 300 to 600 m range. At 350 m depth lowest oxygen values of 3.8 µmol/kg were reached in early December 2014 (Fig. 3b). The massive deepening of the pycnoclines is also reflected by maxima in salinity (34.69 psu) and temperature (10.33°C) in 350 m depth (not shown). The ACE2 shows high meridional velocities of more than 5 cm/s in the upper 450 m

depth.

The lowest SLA and geopotential anomaly of the mooring deployment period was connected to a strong upward displacement of the isopycnals in February/March 2015 (Fig. 3b). The doming of the pycnoclines from January to March 2015 is associated with the typical signature of cyclonic eddies, which uplift colder, less saline and low oxygen waters to shallower depths. Low values in oxygen,

temperature and salinity at 183 m depth of 69.1 µmol/kg, 11.7°C and 34.54 psu in early February 2015 were related to a cyclonic eddy (CE) with a radius of 71 km. Water properties of the CE may be associated with the Eastern South Pacific Intermediate Water which is transported by equatorward surface currents (Chaigneau et al., 2011).

According to the SLA satellite maps the centre of the MWE passed north of the mooring. The centre of

the westward propagating ACE1 and the ACE2 passed the Stratus mooring only 14 km respectively 17 km south of it, hence the Stratus measurements were close to the centre of these two eddies (Supplement Movie M1). However, satellite data show that the mooring captured only the northern segment of the ACE1 (Fig. 4b), therefore the radius of 28 km determined from measurements at the



Stratus mooring is small in comparison to a mean radius of 40 km from satellite maps (Fig. 5a). Oxygen anomalies from January to March 2015 are related to by two consecutive cyclonic eddies explaining the long lasting and strong anomaly. The first eddy (CE) passed the mooring 43 km north of it and then merged with a second cyclonic eddy and passed to the south of the mooring.

Therefore, eddy events lead to a strong signal in water mass properties up to 196 (40) µmol/kg in oxygen, 0.84 (0.18) psu in salinity and 6.0 (1.9) °C in temperature in 183 m (350 m) depth. The oxygen time series at 107 m depth (Supplement Fig. S1) does not show larger anomalies from the mean values in March and late August, at the time the Stratus mooring near sea surface temperature signal showed the maximum and minimum of a seasonal signal (Colbo and Weller, 2007; their Fig. 3). Hence, the

maxima and minima described above for the Stratus time series at 183 m and 350 m are clearly related to eddies and not related to the seasonal signal in the upper layer of the open ocean.

**4.3 Net transport of heat, salt, and oxygen via eddies in the ETSP**

The question on how much anomalous water properties an eddy is able to trap and transport into the

open ocean depends on the relation between swirl velocity and propagation velocity. The MWE and the ACE2 have a similar propagation velocity of 4.3 and 4.2 cm s$^{-1}$, respectively. The CE propagates fastest (6 cm s$^{-1}$) and the ACE1 propagates slowest (3.2 cm s$^{-1}$) (Fig. 6a), which fits well to the mean westward propagation speed of 3-6 cm s$^{-1}$ and 4.3 cm s$^{-1}$ estimated for eddies in the region off Peru (Chaigneau et al., 2008; 2011).

The observed swirl velocity at Stratus mooring is in accordance with other values measured in the ETSP (Chaigneau et al., 2011; Stramma et al., 2013; 2014). All anticyclones (MWE, ACE1 and ACE2) show high rotation values in the upper 200 m depth with maximum velocities of 11, 13 and 17 cm s$^{-1}$ respectively (Fig. 6a), which is between the values of mean anticyclonic eddies (9 cm s$^{-1}$, Chaigneau et al., 2011) and a strong relatively young anticyclonic coastal mode water eddy (35 cm s$^{-1}$, Stramma et

al., 2013). Further, the stronger swirl velocity of ACE2 agrees very well with an anticyclonic eddy, which passed the Stratus mooring during September to December 2011 at about the same season as for ACE2 (19 cm s$^{-1}$, Stramma et al., 2014). Typically for anticyclones, largest velocities occur in the upper

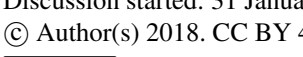


250 m depth, whereas the MWE shows weak rotation in the near surface and a deeper core instead. Below 250 m depth, the swirl velocity of ACE1 is significantly weaker than the swirl velocity of ACE2. Nonetheless, due to the much slower propagation velocity of ACE1 the fluid stays trapped within the eddy (U/c >1) leading to a deep vertical extent of both anticyclones of 504 m (ACE1) and 523 m

(ACE2) as described for mean anticyclones in the ETSP (Chaigneau et al., 2011).

Within the eddy boundaries of the two anticyclones (ACE1 and ACE2), positive anomalies of temperature and salinity were observed between 100 and 600 m depth (Fig. 6b, c). Between 150 and 200 m depth maximum anomalies of 2.1°C (ACE1) and 1.5°C (ACE2) in temperature and 0.35 psu (ACE1) and 0.24 psu (ACE2) in salinity were measured which are significantly higher than described

for the mean of a composite of anticyclones (0.8°C, 0.08 psu; Chaigneau et al., 2011). Due to the uplift (depression) of isopycnals above (below) 200 m depth the MWE shows negative (positive) anomalies in temperature and salinity between 50 and 200 m (below 200 m) depth, which are weak in comparison to the other two anticyclones.

Oxygen shows a mainly negative anomaly below 220 and 280 m depth respectively within the

anticyclones MWE, AC1 and AC2. Both, the MWE and the ACE2 are having their largest negative anomalies in 250 m depth and a second minimum in 450 m, which is just above and below the core of the OMZ indicating the transport of low oxygenated water masses from a region with a larger vertical expansion of the OMZ (Fig. 6d). In the core depth of the OMZ at 350 m depth only weak negative oxygen anomalies are possible as the oxygen content is already low, but still the passage of the stronger

anticyclone ACE2 results in oxygen decrease by 14 μmol/kg.

Water mass anomalies within the MWE lead to an available heat, salt and oxygen anomaly (AHA, ASA, AOA) of 0.8 x $10^{18}$ J, -3.6 x$10^{10}$ kg and -3.5 x$10^{16}$ μmol. These values are about five times smaller in comparison to a mode-water eddy that was also measured at the Stratus mooring in February/March 2012 (Stramma et al., 2014; Tab. 1). ASA is even negative in 2015 due to the strong

doming in the upper 200 m. As both mode-water eddies have about the same propagation speed and volume, the differences of water mass properties point towards seasonal or interannual variations of the water mass characteristics during their formation. The MWE is generated in February, when upwelling-favorable alongshore winds weaken and SST increases (Gutiérrez et al., 2011), whereas the mode-water



eddy observed by Stramma et al. (2014) is generated in April, when the PCUC, which transports oxygen-deficient ESSW (Hormazabal et al., 2013), has its poleward maximum (Shaffer et al., 1999; Penven et al., 2005; Chaigneau et al., 2013). Additionally, the mode-water eddy observed by Stramma et al. (2014) was generated in year 2011, which is considered as a La Niña-period, when generally

lower oxygen and higher salinity values exist in the upper 100 m depth (Stramma et al., 2016) leading to higher anomalies of salt and oxygen of the MWE observed by Stramma et al. (2014) in comparison to the actual MWE. In addition, the MWE passed to the north of the Stratus mooring during its deployment, hence the method to define the fully MWE parameter might lead to higher deviations to the real eddy parameters.

The volume of ACE2 (4.6 $\times 10^{12}$ m$^3$) is in agreement to the mean anticyclones (Chaigneau et al., 2011) and the open-ocean anticyclone (Stramma et al., 2013) but three times larger than the weaker ACE1, which is partly due to the underestimated radius (Tab. 1). The AHA, ASA, and AOA of ACE2 are 7.6 x $10^{18}$ J, 23.9 $\times 10^{10}$ kg and -3.6 $\times 10^{16}$ µmol and therefore far greater than the AHA, ASA and AOA of ACE1 (1.8 x $10^{18}$ J, 5.4 $\times 10^{10}$ kg, -0.02 $\times 10^{16}$ µmol). The weak negative AOA of ACE1 result from the

strong and positive oxygen anomaly in the upper 280 m depth. Strong differences between the results for ACE1 and ACE2 are of course due to the higher volume of the ACE2 but might also reflect the conditions at different seasons during the formation of the eddies leading to varying water mass properties. ACE1 (ACE2) is generated in austral summer (winter) when upwelling-favourable winds weaken (strengthen). Estimations of AHA and ASA within ACE2 match the mean values of Chaigneau

et al. (2011). In comparison to the open-ocean eddy (Stramma et al., 2013) the AHA of the ACE2 is twice as high, but the AOA is only half as much (Tab. 1).

The cyclonic eddy in March 2015 (CE) has maximum velocities of 14 cm s$^{-1}$ in 50 m depth (Fig. 6a). Due to the high translation speed of the CE, the vertical extent of 176 m depth is much shallower than the vertical extent of the anticyclones, which is consistent with the mean cyclones (Chaigneau et al.,

2015). The mean temperature shows a pronounced negative anomaly within the eddy boundaries between 70 m and 430 m depth with a maximum anomaly of -2.3°C in 160 m (Fig. 6b) resulting in a negative AHA of -9.4 x $10^{18}$ J. The salinity is negative between 40 m and 220 m depth with a maximum anomaly of -0.32 psu in 160 m depth (Fig. 6c). This results in an extremely large negative ASA of -42.8



$x10^{10}$ kg. Oxygen shows a strong negative anomaly in the upper 320 m having a maximum anomaly of -69 µmol/kg in 180 m depth (Fig. 6d) due to the uplift of the main thermocline leading to a negative AOA of -6.5 $x10^{16}$ µmol for the 107 to 176 m depth layer. Although the volume of the CE is in good agreement with the mean values of Chaigneau et al. (2011), the estimated AHA and ASA are much

higher (Tab. 1), which is likely due to strong seasonal variations during the generation of the CE. The CE is formed in austral winter off Peru when coastal alongshore winds intensify leading to an enhanced upwelling of cold as well as nutrient-rich and oxygen-poor water due to high biological production. Additionally, equatorward surface currents, which transport the relatively cold and fresh Eastern South Pacific Intermediate Water, are strongest during austral winter (Gunther, 1936).

Fluxes of mass, heat, salinity, and oxygen are estimated from the volume, AHA, ASA, and AOA (Tab. 1) for the period in which the MWE, ACE2, and CE cross the 85°W longitude at the Stratus mooring. The ratio of the heat fluxes of the three different eddies mirror the differences between volume, AHA, ASA, and AOA of the respective eddies because of the similar duration of the passages of the eddies. Transports of mass anomaly for the CE, ACE2 and MWE are again similar and range between 1.1 Sv

for the CE to 1.8 Sv for ACE2 (Tab. 2). Due to the local conditions and different water masses during its formation, the CE shows a strong negative transport of heat ($-4.0$ x $10^{12}$ W) and salt ($-18.1$ x $10^4$ kg s$^{-1}$) across the mooring, whereas the ACE2 transports the highest positive amount of heat ($3.0$ x $10^{12}$ W) and salt ($9.5$ x $10^4$ kg s$^{-1}$) per year. Whereas the transport of heat and salt of the MWE is relatively small in comparison to the CE and ACE2, the transport of low oxygen water of $-1.8$ x $10^{10}$ µmol s$^{-1}$ is in the

same range as the CE and ACE2 due to a thick lens of low-oxygen water within the MWE.

Available anomalies of heat, salt, and oxygen of cyclonic and anticyclonic eddies gained from the Stratus mooring and from the literature (Table 1) are now used to estimate the relative eddy contribution to fluxes of mass, heat, salt, and oxygen in the ETSP. By multiplying the amount of AHA, ASA, and AOA of the composite eddies with the number of eddies dissipating per year in a given area

(corresponding to a flux divergence) mean heat (in W m$^{-2}$), salt (in kg s$^{-1}$ m$^{-2}$) and oxygen release (µmol s$^{-1}$ m$^{-2}$) were calculated. We define an area reaching in north-south direction from 10°-22°S. The transition area is bordered in the east by the Peruvian and Chilean coast and in the west by the longitude of the Stratus mooring (86°W) corresponding to a size of ~2 x $10^6$ km$^2$. Based on satellite



measurements 228 eddies per year are generated in this area east of 86°W from which 111.7 are cyclones and 116.3 are anticyclones. 16.3 cyclones, and 13.9 anticyclones and mode-water eddies propagate into the area west of the Stratus mooring meaning that 95.4 of the cyclones and 102.4 of the anticyclones and mode-water eddies have dissipated and therefore released a certain amount of heat,

salt, and oxygen in the transition zone. Based on the mean of AHA, ASA, and AOA for the composite eddies the mean heat (salt, oxygen) release per year and m$^2$ is -2.4 x 10$^{13}$ W m$^{-2}$ (-8.9 x 10$^5$ kg s$^{-1}$ m$^{-2}$, -2.0 x 10$^{11}$ μmol kg$^{-1}$ s$^{-1}$ m$^{-2}$) for cyclones and 1.6 x 10$^{13}$ W m$^{-2}$  (5.0 x 10$^5$ kg s$^{-1}$ m$^{-2}$, -2.1 x 10$^{11}$ μmol kg$^{-1}$ s$^{-1}$ m$^{-2}$) for anticyclones and mode-water eddies.

Heat and salt fluxes across the Stratus mooring as well as for the ETSP reveal an imbalance between

anticyclones and cyclones which are due to a stronger transport of colder and fresher water within cyclones from the coast off Peru and Chile into the open ocean. Both types of eddies show negative oxygen fluxes meaning that anticyclones and cyclones transport less oxygenated water into the open ocean and therefore have an impact on the maintenance and size of the OMZ in the ETSP, which is also confirmed by models (I. Frenger, pers. communication).

**4.4 Properties of the observed eddies MWE, ACE1/2 and CE during their lifetime**

With the help of satellite data the four eddies (MWE, ACE1/2 and CE) could be identified and followed from areas near the Peruvian and off the Chilean coast to the areas of dissipation westwards of the Stratus mooring in the open ocean. The trajectories of the three anticyclonic eddies (MWE, ACE1/2) were extrapolated to the formation regions near the coast between 21°S and 23°S (Fig. 1). The CE

formed during end of July 2014 off the Peruvian coast during the winter season when upwelling is usually strong and decayed in mid-March 2015 after propagating 1200 km in more than seven months. Both the MWE and the ACE1 started in February 2013 off the Chilean coast during the summer season with usually low upwelling. The MWE can be followed for about two years till March 2015 propagating 2880 km. The ACE1 is tracked for 620 day until it decayed end of November 2014 after propagating

1750 km. The ACE2, which was generated during the upwelling season end of winter in September 2013 and decayed in June 2015, propagated westward for 2350 km in 650 days.





As expected from the polarity depending meridional deflection of all eddies (anticyclones – equatorward, cyclones - poleward) also the individual pathways of the ACE1 and ACE2 show a north-westward direction whereas the CE migrates more south-westwards (Fig. 1a). Note, that the MWE shows no clear meridional deflection on the way to the west.

Anticyclonic eddies (MWE, ACE1/2) are associated with a positive SLA, wherein ACE2 shows the strongest mean elevation of all anticyclonic eddies of 8 cm (Fig. 4c) and cyclonic eddies are identified by a negative SLA, wherein the CE shows a mean minimum SLA of -2 cm in the centre of the eddy (Fig. 4d). Nevertheless, the CE showed the largest SLA differences at the Stratus mooring (Fig. 3a). In general, mode-water eddies are difficult to detect by satellite altimetry due to a relatively weak velocity

at the near-surface (Fig. 6a) which is generated by the typical distribution of the isopycnals. Therefore, it is noteworthy that the MWE has a stronger SLA (7 cm) than the relatively weak ACE1 (6 cm). The SLA of the MWE indicates higher variability during its lifespan than the other eddies (Fig. 5e). The maximum SLA of the anticyclonics is obtained during their mid-age, whereas the SLA of the CE decreases after the very beginning.

Due to the uplift of the seasonal pycnocline in both eddies, MWE as well as CE, cold and nutrient rich water is upwelled into the euphotic zone leading to enhanced biological production, which is reflected by negative SST anomalies of -0.04°C (MWE, Fig. 4e) and -0.08°C (CE, Fig. 4h) and high chlorophyll production of 0.19 mg m$^{-3}$ (CE, Fig. 4l). Surprisingly, the mean SST (Chl) of the ACE1 and ACE2 are negative (high) and around zero, respectively, as one would expect positive (low) SST (Chl) anomalies

due to the depression of the thermoclines. The development of the SST predominantly shows negative anomalies with short periods of positive anomalies for the anticyclonics (Fig. 5f).

The development of further eddy properties (radius [km], rotating velocity [m s$^{-1}$], nonlinearity parameter and Rossby Number) during the normalized lifespan are shown in Figure 5 indicating that the observed eddies pass the Stratus mooring during their mid-age (MWE and ACE2) and during the end of

their lifetime (ACE1 and CE). The anticyclonics ACE1/2 have their maximum radius during the last third of their lifetime, whereas the development of the radius of the MWE is symmetric and the radius of the CE increases during the first third of its lifetime (Fig. 5a).



Water mass anomalies can only be preserved within an eddy if the feature is nonlinear and maintains its coherent structure. During their full lifetime the nonlinear parameter U / c > 1 for all eddies confirming the coherent feature (Fig. 5c). MWE and CE show stronger fluctuations of the nonlinear parameter than the ACE1/2, which mirrors the higher variability of the swirl velocity of both eddies (Fig. 5b).

All eddies indicate a Rossby number $R_o$ <1 describing the typical scale for mesoscale eddies (Fig. 5d). The mean life cycle of an eddy consists of a growth and decaying phase, both lasting about 20% of its lifetime and a stable phase in between (Liu et al., 2012; Frenger et al., 2015; Samelson et al., 2014), which is not consistent with our observations showing Rossby numbers of less than 0.1 for ACE1 and ACE2 reflecting a stable phase over 90 % of the lifetime of the eddies. In contrast to this, MWE and CE

show a longer growing phase of 30 % and also a longer decaying phase of 40 %, where the stable phase with a Rossby number of less than 0.1 remains short.

**4.5 Observations of Argo floats within the eddy-core of ACE2 during its mid-age**

ACE2 has been tracked via SLA from February 2014 on, passed the Stratus mooring in November 2014

and decayed in June 2015. During this period, four floats (Fig. 7) were captured within the ACE2 during its westward propagation at different times providing information about different stages of the eddy. The first float (#6900532) was trapped in the period from mid-April to mid-May 2014 at 77.1°W, 19.9°S (supplement: movie M1). The eddy shows the core at about isopycnal 26.4 kg m$^{-3}$ (~180 m depth) with extremely low oxygen of less than 4 μmol kg$^{-1}$ between 150 and 400 m depth (Fig. 7a) and

enhanced salinity of more than 34.8 psu in the upper 240 m depth (Fig. 8a). The warm, salty and oxygen-depleted water mass of the core of the ACE2 coincides with the water mass of the likely formation region of the ACE2 obtained from the MIMOC climatology (Fig. 9a, b) reflecting the characteristics of the oxygen-depleted ESSW. The ESSW is carried poleward by the secondary southern subsurface countercurrent (Montes et al., 2014), feeds the subsurface PCUC (Hormazabal et al., 2013)

and is then transported along the Peruvian and Chilean coast where anticyclonic eddies are likely generated  (Chaigneau et al., 2011).

In September 2014, about four months later and more than 6° further west at 83.4°W, 19.4°S, a second float (#6900530) was trapped in the same eddy ACE2 (supplement: movie M1). The core was still



located at isopycnal 26.4 kg m$^{-3}$ (~230 m depth) showing minimum oxygen values of less than 4 µmol kg$^{-1}$ (Fig. 7b) and maximum salinity of more than 34.8 psu (Fig. 8b) between 200 and 280 m depth. However, the vertical extent of anomaly high salinity and anomaly low oxygen has decreased (Fig. 9a, b) which is likely due to lateral mixing by turbulent diffusion at the boundary of the eddy. Mixing

mostly takes place above the core of the eddy between the density layers $\sigma_\theta$=25.7 and $\sigma_\theta$=26.3 kg m$^{-3}$ (supplement Fig. S2) with largest changes in oxygen (0.5 µmol kg$^{-1}$ day$^{-1}$), temperature (-0.007°C day$^{-1}$), and salinity (-0.002 psu day$^{-1}$) at about $\sigma_\theta$=26.0 kg m$^{-3}$. This density level corresponds to a depth between 100 and 170 m, where high velocities exist within the ACE2 (Fig. 6a), which are essential to keep up the coherent structure and therefore should inhibit lateral mixing.

Shortly after, in October 2014, a third float (#6900529) stayed in the ACE2 at about 82.8°W, 20.4°S. The strongest water property anomaly is now located at isopycnal 26.6 kg m$^{-3}$ (~310 m depth) showing minimum oxygen of less than 8 µmol kg$^{-1}$ (Fig. 7c). The salinity anomaly transported within the eddy has declined furthermore to about 34.65 psu (Fig. 8c). The development of the water mass properties within the eddy points towards mixing along density surfaces between $\sigma_\theta$=26.0 kg m$^{-3}$ (~190 m) and

$\sigma_\theta$=26.5 kg m$^{-3}$ (280 m). The changes are strongest above the core of the eddy at about $\sigma_\theta$=26.3 kg m$^{-3}$ (~205 m to 240 m) showing the mixing of oxygen-rich (2.8 µmol kg$^{-1}$ day$^{-1}$), colder (-0.07°C day$^{-1}$) as well as fresher water (-0.017 psu day$^{-1}$) into the ACE2 (supplement Fig. S2).

Decreased anomalies might also be related to the fact that the float did not capture the eddy centre as it was located in the southeast rim of the eddy (Fig. 4c). Whereas the salinity measurements of the float

differ from those of the Stratus measurements obtained during the passage of the ACE2 (Fig. 9b), the oxygen anomaly transported in the core agrees well (Fig. 9a).

After the ACE2 has passed the Stratus mooring in November 2014, the last of the four floats (#6900527) was trapped at the southern rim of the eddy (Fig. 4c) from December 2014 to January 2015 at about 86.4°W, 20.4°S. The eddy core is still clearly visible, although water mass properties within the

core of the ACE2 has further changed (oxygen > 10 µmol kg$^{-1}$, Fig. 7d) and the vertical extent of the eddy has declined. Mixing of slightly oxygen-richer water can be observed in the whole eddy (supplement Fig. S2a), whereas warmer and more saline water is entrained in the upper part of the eddy





($\sigma_0 > 26.3$ kg m$^{-3} \cong 240$ m depth) and colder and fresher water below. These changes might also be due to the location of the float outside the eddy boundary.

## 5. Discussion and conclusion

The ETSP is known for its high eddy frequency (Chaigneau et al., 2008). There is still limited
knowledge in this region about the dynamics of eddies especially on their effective transport and their dissipation. In this study the activity of three different types of eddies (mode water, anticyclonic, and cyclonic eddy) during their westward propagation was investigated from the formation area in the upwelling area off Peru and Chile into the open ocean. The focus was on the development of the eddies, seasonal conditions during their formation, and the change of water mass properties transported within
the isolated eddies using a broad range of observational data such as SLA, SST, and Chl from satellites as well as hydrographic data and oxygen from the Stratus mooring and from Argo floats.

Available heat, salt, and oxygen anomalies could be computed for the investigated eddies. Generally, heat and salt anomalies transported within eddies are positive for anticyclones and negative for cyclones and might be compensated as they are of about the same amount (Chaigneau et al., 2011). In contrast, in
this study negative anomalies of the water mass properties within the observed cyclonic eddy are too high for heat and salt (-9.4 x10$^{18}$ J, -42.8 x10$^{10}$ kg) as they could be balanced by positive anomalies of heat and salt transported within the anticyclonic (7.6 x10$^{18}$ J, 23.9 x10$^{10}$ kg) and the mode water eddy (0.8 x10$^{18}$ J, -3.6 x10$^{10}$ kg). Therefore, seasonal variability such as fluctuation of alongshore upwelling-favourable winds off Peru and Chile as well as interannual variability such as El Niño/La Niña have an
impact on the water mass properties trapped and transported within eddies from the coast of Peru and Chile into the open ocean reflecting the high variability of AHA and AHA. The AOA is negative for all types of eddies (MWE: -3.5 x10$^{16}$ µmol; ACE2: -3.6 5 x10$^{16}$ µmol; CE: -6.5 5 x10$^{16}$ µmol), whereby the transport of oxygen-low water from the upwelling region into the open ocean is more surface intensified due to the shallow structure of the cyclonic eddies.

Satellite-based estimate of the surface-layer eddy heat flux divergence, while large in coastal regions, is small when averaged over the southeast Pacific Ocean, suggesting that eddies do not substantially



contribute to cooling the surface layer in this region (Holte et al., 2013). In this study, the release of fluxes of heat (cyclones: -2.4 x $10^{13}$ W m$^{-2}$; anticyclones: 1.6 x $10^{13}$ W m$^{-2}$) and salt (-8.9 x $10^5$ kg s$^{-1}$ m$^{-2}$; 5.0 x $10^5$ kg s$^{-1}$ m$^{-2}$) estimated from eddies dissipating in the ETSP confirms the discrepancy between different types of eddies leading to a net transport of colder and fresher water from the formation

regions off Peru and Chile into the open ocean. In contrast, all three types of eddies show a negative oxygen flux of -2.0 x $10^{11}$ µmol kg$^{-1}$ s$^{-1}$ m$^{-2}$ for cyclones and -2.1 x $10^{11}$ µmol kg$^{-1}$ s$^{-1}$ m$^{-2}$ for anticyclones and mode-water eddies pointing towards an active role of eddies in maintaining and shaping the OMZ.

For the Atlantic Ocean the low-oxygen eddy cores has been attributed to high productivity in the

surface (Schütte et al., 2016b), enhanced respiration of sinking organic material at subsurface depth (Fiedler et al., 2016) and a strong isolation of the eddy core (Karstensen et al., 2017). An anticyclonic mode water eddy observed at the Stratus mooring in February/March 2012 indicated high primary production just below the mixed layer (Stramma et al., 2014). According to a global investigation the eastern South Pacific off Peru and Chile seems to have the highest amount of MWEs, which are also

deep reaching compared to other regions (Zhang et al., 2017). Nevertheless, in the mooring deployment period 2014/2015 only one MWE crossed to the north of the mooring and the results have to be regarded with caution. Even though the AHA and ASA of the MWE are small in comparison to both the anticyclonic eddies and the cyclonic eddy the transport of low oxygen water is in the same range as the other eddies due to the typical thick lens of low oxygen water within mode water eddies.

From a combination of satellite data and Argo profiles long-lived eddies (lifetime longer than 30 days) in the Peru-Chile upwelling system 55% of the sampled anticyclonic eddies had subsurface-intensified maximum temperature and salinity anomalies below the seasonal pycnocline, whereas 88% of the cyclonic eddies are surface intensified (Pegliasco et al., 2015). The 55% subsurface-intensified anticyclonic eddies represent mode water eddies while the 45% surface intensified anticyclones are

'regular' anticyclonic eddies. Eddy generation off the coast between 8° and 24°S peak in austral summer/spring, which agrees with the strengthening of the PCUC and the possible mechanism of the generation of eddies due to instabilities. However, this is not in agreement with model simulations showing an eddy-induced offshore transport off Peru that peaks in austral spring (winter) at the southern



(northern) boundary of the OMZ (Vergara et al., 2016). As the PCUC also shows strong fluctuations lasting a few days up to a few weeks (Huyer et al., 1991) it might be difficult to determine a seasonal dimension between the PCUC and the rate of eddy formation. El Niño (La Niña) events deepen (shoal) the thermocline and intensify (weaken) the PCUC (e.g. Montes et al., 2011; Combes et al., 2015) and

therefore might play a role for the amount of eddy generation on interannual time scales.

The development of isolated water mass properties was investigated on the basis of four floats trapped within an anticyclonic eddy indicating lateral mixing of all water properties (oxygen, temperature, and salinity). The water mass within the core of the ACE2 shows the typical characteristics of the salty and oxygen-depleted ESSW. The mixing is strongest between the seasonal thermocline and the core of the

eddy at about $\sigma_\theta$=26.3 kg m$^{-3}$ (~205 to 240 m depth) and takes place during the first half of the lifetime of the eddy. Stronger mixing during the first half of the eddy lifetime might be related to a stronger wind curls near the coast in comparison to the open ocean. However, as the changes might also be due to the fact that the floats might be located at different positions within the eddy or near the outer edge these results show be regarded with caution. As floats are carried with the eddies near the outer edge of

the eddy, the floats avoid the core of eddies and hence underestimate the strength of eddies. Especially floats with a parking depth at 400 m stay near the edge of eddies and shift between cyclonic and anticyclonic features. If located in the ETSP at southern boundaries of anticyclones or northern boundaries of cyclones the floats move eastward while floats at northern boundaries of anticyclones or southern boundaries of cyclones move westward (Supplement Movie M1).

The zonal annual mean velocity components for the periods 2011/2012 and 2014/2015 as well as for the period October 2000 to December 2004 are quite different (between -4.6 and +2.5 cm s$^{-1}$), while the meridional components are weaker (between -2.1 and 2.3 cm s$^{-1}$) and more similar (Supplement Fig. S3). In 2011/2012 the flow component is south-eastward between 100 and 450 m depth and in 2014/2015 south-westward in the upper 50 m depth, north-westward at 50 to 250 m depth and mainly

southward at 250 to 600 m depth. The north-westward flow at 50 to 250 m depth in 2014/2015 would fit to a mean South Pacific subtropical gyre circulation however the period 2011/2012 showed mean eastward flow at all depths between 50 and 600 m and also the 2000/2004 period (Colbo and Weller, 2007) eastward flow component at 45 to 235 m show the opposite flow component. Hence, the different



measurement periods at the Stratus mooring do not support the view of a mean north-westward flow of the South Pacific subtropical gyre at the mooring location and other processes dominate the flow field. Observations from the mooring deployment period March 2014 to April 2015 and profiling floats deployed near eddies in March 2014 (Fig. 1) show the importance of eddies in the weak flow region of

the ETSP as mesoscale eddies are a main contributor for water mass distribution. High anomalies of heat, salt, and oxygen transported within eddies into the ETSP suggests the influence of seasonal and/or interannual variability in the formation regions off Peru and Chile. It is necessary to further gain knowledge about the seasonal and interannual variability having an impact on the generation of eddies and on the water mass properties that are trapped within eddies to understand their impact on the

maintenance and shape of the OMZ.

**A Supplement is related to this article.**

*Author contributions.* R. Czeschel, F. Schütte, and L. Stramma conceived the study. R. Czeschel handled the float data. F. Schütte collected and interpreted the satellite data set. R.A. Weller is responsible for the Stratus ORS mooring measurements and the data processing. All authors discussed, wrote and modified the manuscript.

*Acknowledgements.* Financial support was received through Woods Hole Oceanographic Institution (R.A.W.) and the GEOMAR (R.C., L.S. and F.S.). The Stratus Ocean Reference Station is supported by the National Oceanic and Atmospheric Administration (NOAA) Climate Observation Program (NA09AR4320129, OAA CPO FundRef number (100007298)). This work is a contribution of the Deutsche Forschungsgemeinschaft (DFG) supported project "Sonderforschungsbereich 754: Climate-

Biogeochemistry Interactions in the Tropical Ocean" (http://www.sfb754.de). The Copernicus Marine and Environment Monitoring Service (CMEMS) (http://marine.copernicus.eu) has taken over the whole processing and distribution of the products formerly distributed by AVISO with no changes in the scientific content.



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





**Table 1.** Properties and available heat, salt and oxygen anomalies (AHA, ASA, AOA) of one mode-water eddy (MWE), two anticyclones (AE1 and AE2) and one cyclonic eddy (CE) measured at the Stratus mooring in 2014/2015 within the vertical layer of the coherent structure in comparison to measurements in February/March 2012 (Stramma et al., 2014 (STR14), mode-water eddy) at the Stratus mooring, at 83°50'W, 16°45'S in November 2012 (Stramma et al., 2013 (STR13), anticyclone) and mean values for 10°S-20°S relative to a mean climatology (Chaigneau et al., 2011 (CH11), anticyclones and cyclones). Based on instruments available the vertical extent for heat and salt computations (TS) and oxygen (OX) differs. It is important to note that the radius of the ACE1 might be underestimated.

| | Mode-water eddies | | Anticyclones | | | | Cyclones | |
|---|---|---|---|---|---|---|---|---|
| | MWE | STR14 | ACE1 | ACE2 | STR13 | CH11 | CE | CH11 |
| Lifetime (days) | 768 | 315 | 624 | 650 | | | 229 | |
| Propagation (cm s$^{-1}$) | 4.3 | 4 | 3.2 | 4.2 | 4.8 | 4.3 | 6 | 4.3 |
| Radius (km) | 43 | 38 | (28) | 53 | 49 | 58 | 71 | 62 |
| Vertical extent (m) TS/OX | 43-620/ 107-600 | 45-600 | 13-504/ 107-504 | 13-523/ 107-523 | 0-600 | 0-450 | 13-176/ 107-176 | 0-200 |
| Volume (x10$^{12}$ m$^3$) TS/OX | 3.4/2.9 | 2.5 | (1.3/1) | 4.6/3.8 | 4.7 | 4.9 | 2.7/1.1 | 2.6 |
| AHA (x10$^{18}$ J) | 0.8 | 5.8 | (1.8) | 7.6 | 3.7 | 6.5 | -9.4 | -5.9 |
| ASA (x10$^{10}$ kg) | -3.6 | 19.3 | (5.4) | 23.9 | 18.7 | 17.4 | -42.8 | -14.7 |
| AOA (x10$^{16}$ μmol) | -3.5 | -10.5 | (-0.02) | -3.6 | -7.6 | -/- | -6.5 | -/- |





**Table 2.** Mass, heat, salt and oxygen transport of MWE, ACE2, CE across 85°W of the Stratus mooring 2014/2015.

|  | MWE | ACE2 | CE |
|---|---|---|---|
| Mass transport (Sv) | 1.7 | 1.8 | 1.1 |
| Heat transport (x $10^{12}$ Watt) | 0.4 | 3.0 | -4.0 |
| Salt transport (x $10^{4}$ kg s$^{-1}$) | -1.8 | 9.5 | -18.1 |
| Oxygen transport (x $10^{10}$ μmol s$^{-1}$) | -1.8 | -1.4 | -2.7 |



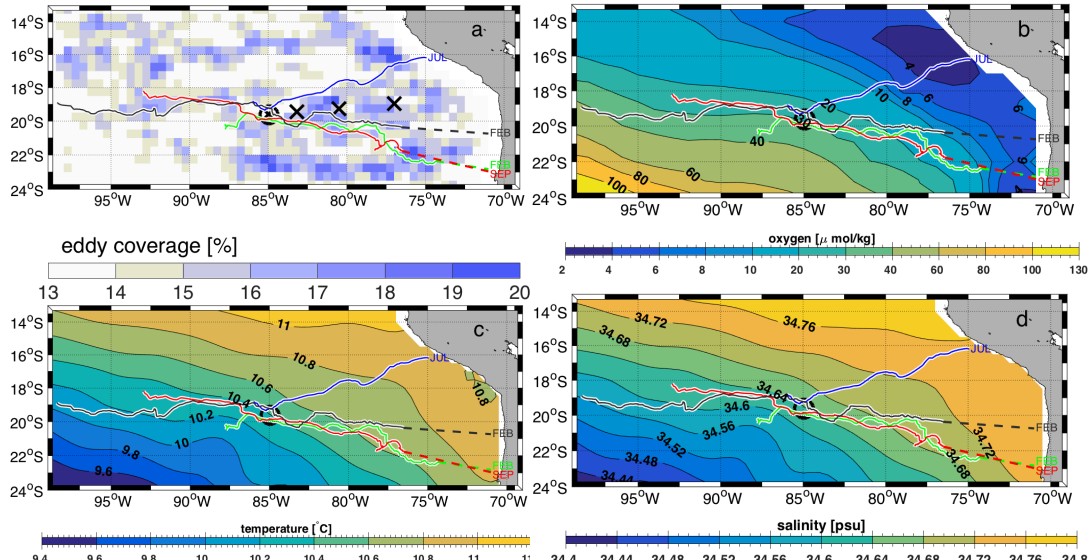

**Figure 1.** a) Percentage of eddy coverage determined from Aviso Sea Level Anomaly (SLA) during the period from 1993 to 2015. The mean distribution of b) oxygen, c) temperature and d) salinity on density surface 26.6 kg m$^{-3}$ derived from MIMOC climatology. The black Xs show the location of the float deployments, the westernmost X with the circle is also the location of the Stratus mooring. The green, red and black lines represent trajectories of the anticyclones (ACE1, ACE2) and the anticyclonic mode water eddy (MWE), whereas the blue line is associated to the trajectory of a cyclone (CE). Dashed lines show the extrapolated tracks to the formation regions and the estimated time of formation. All of these eddies have crossed close to the mooring position and are examined in more detail.




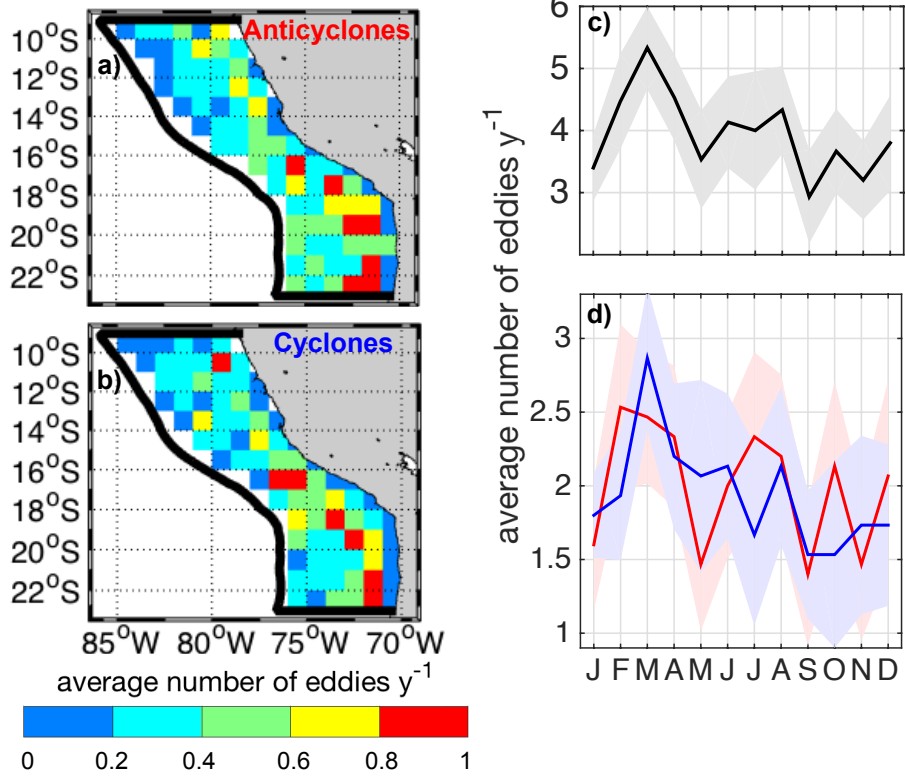

**Figure 2**: Number of a) anticyclones and b) cyclones generated in 1° x 1° boxes (colors) between 1993 and 2015 closer than 600 km off the coast (coastal region). Seasonal cycle of the number of all eddies (black line), anticyclones (red line) and cyclones (blue line) generated in the coastal region are shown in
5    c) and d).



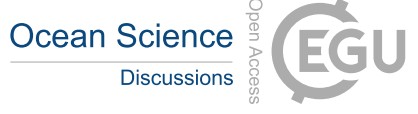

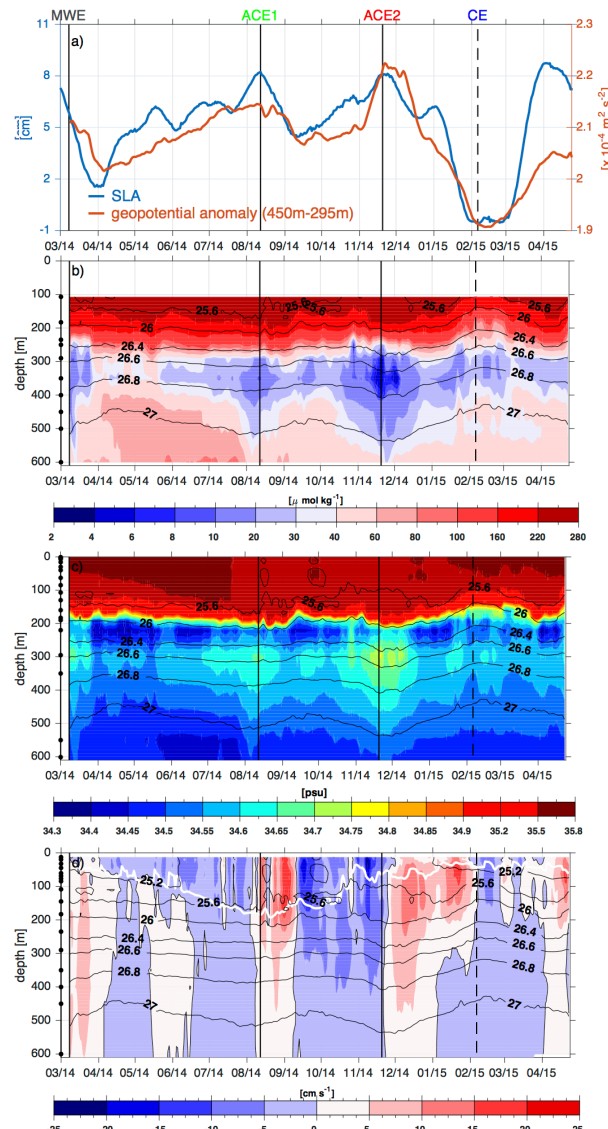



**Figure 3.** Time series of the Stratus mooring (19°37'S, 84°57'W) for the deployment period 8 March 2014 to 25 April 2015 for (a) weekly delayed, high-pass filtered sea surface height anomaly (in cm; blue curve) and geopotential anomaly between 450 and 295 m depth in $m^2 s^{-2}$ (orange curve), (b) oxygen in $\mu mol\ kg^{-1}$, (c) salinity and (d) the meridional velocity component in $cm\ s^{-1}$. The white curve in (d) is the mixed layer depth defined for the depth where the potential density anomaly is 0.125 $kg\ m^{-3}$ larger than at the surface. The black dots on the vertical line at the left mark the depths of the used oxygen (b), conductivity (c) and velocity (c) sensors and the black contour lines are selected density contours. Black solid (dashed) lines show the date of the passages of the anticyclonic (cyclonic) eddies.



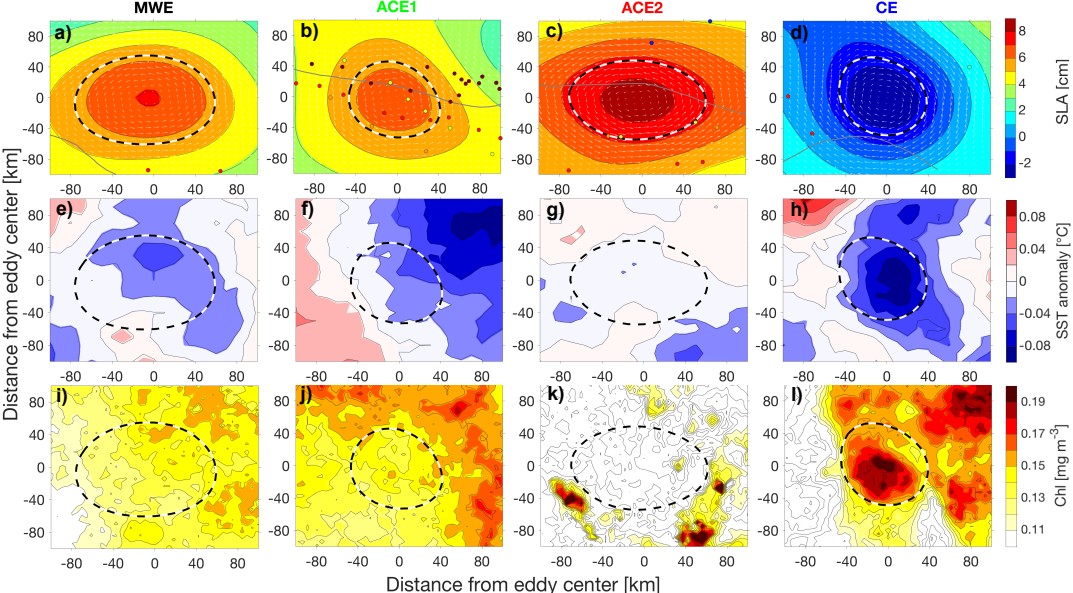

**Figure 4.** Composite of the MWE (a, e, i), ACE1 (b, f, j), ACE2 (c, g, k) and CE (d, h, l) surface signatures for SLA, SST anomaly, and Chl. The dashed black and white line is the eddy boundary, defined as the streamline of strongest velocity. The grey line in a) and d) is the position of the Stratus mooring during the eddy passage.



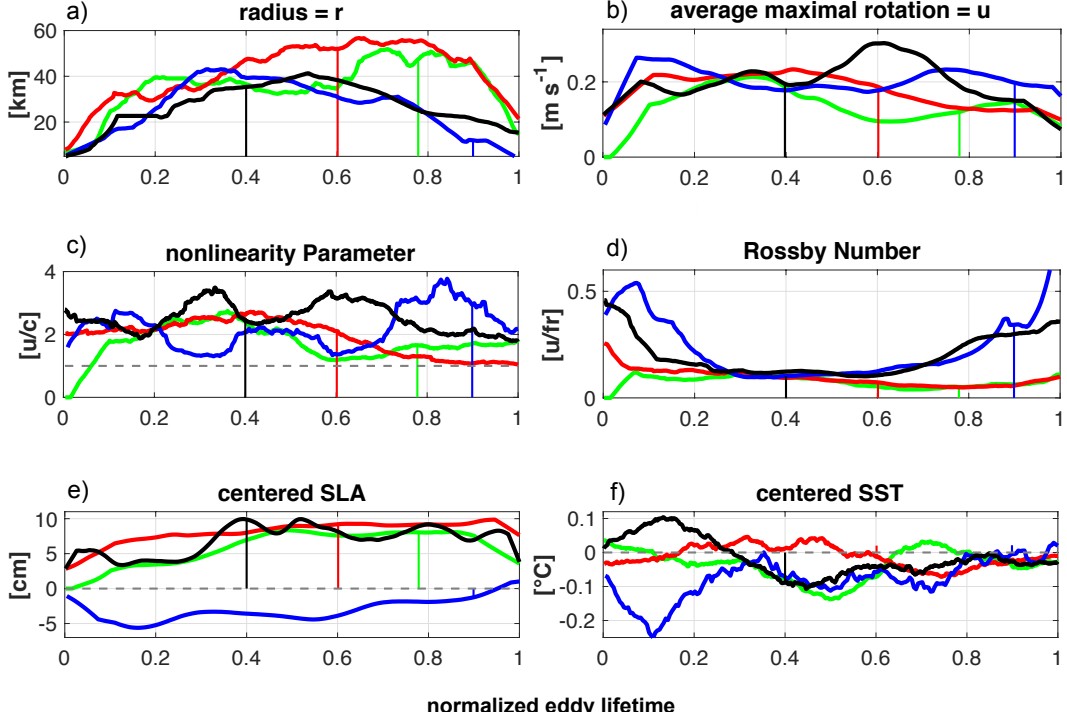

**Figure 5.** a) Eddy radius, b) averaged maximum rotation velocity, c) nonlinearity parameter (u/c), d) Rossby number (U/f*r), e) centred SLA, and f) centered SST anomaly against normalized eddy lifetime of the ACE1 (green), ACE2 (red), CE (blue), and MWE (black). The coloured solid lines mark the passage at the Stratus mooring of the corresponding eddies.



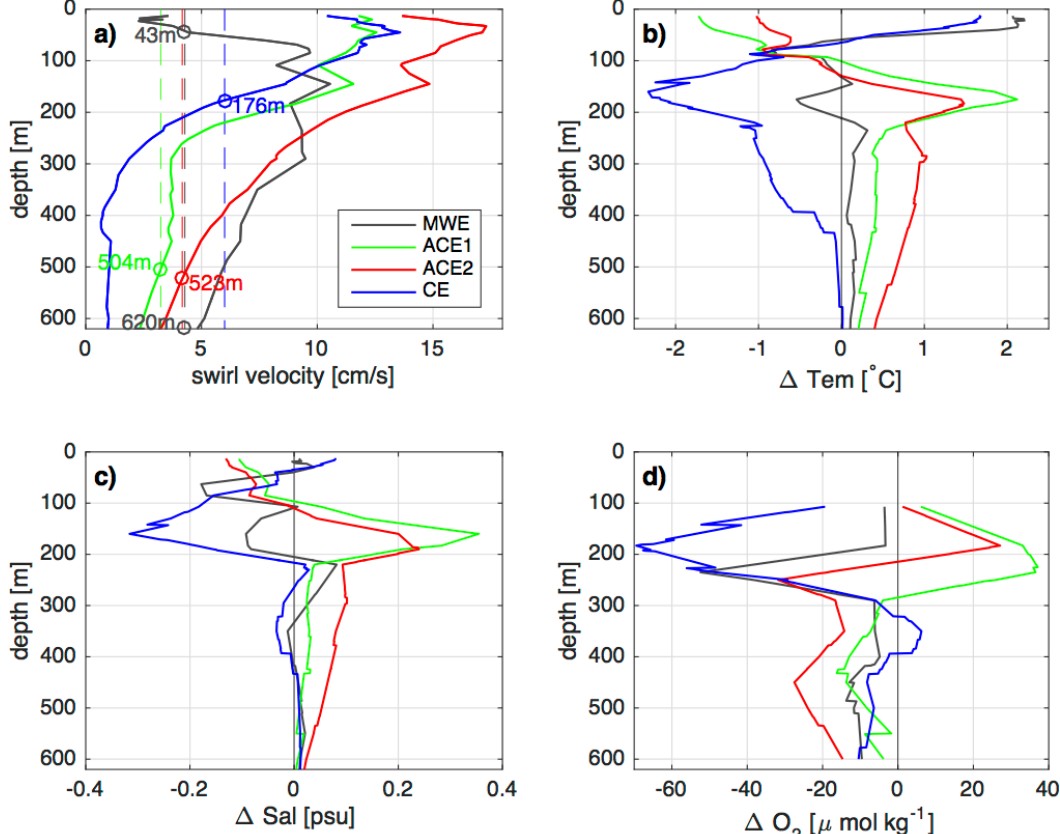

**Figure 6.** a) Swirl velocity vs depth (solid lines), propagation velocity (dashed lines) and vertical extent of the trapped fluid (circles) of three anticyclonic eddies (MWE: grey, ACE1: green, ACE2: red) and a cyclonic eddy (CE: blue). Profiles of anomalies of b) temperature, c) salinity and d) oxygen [μmol kg⁻¹]

5   calculated as the difference between the core of MWE (grey), ACE1 (green), ACE2 (red) and CE (blue) and the 1-year-mean of the Stratus mooring (10 April 2014 - 9 April 2015).





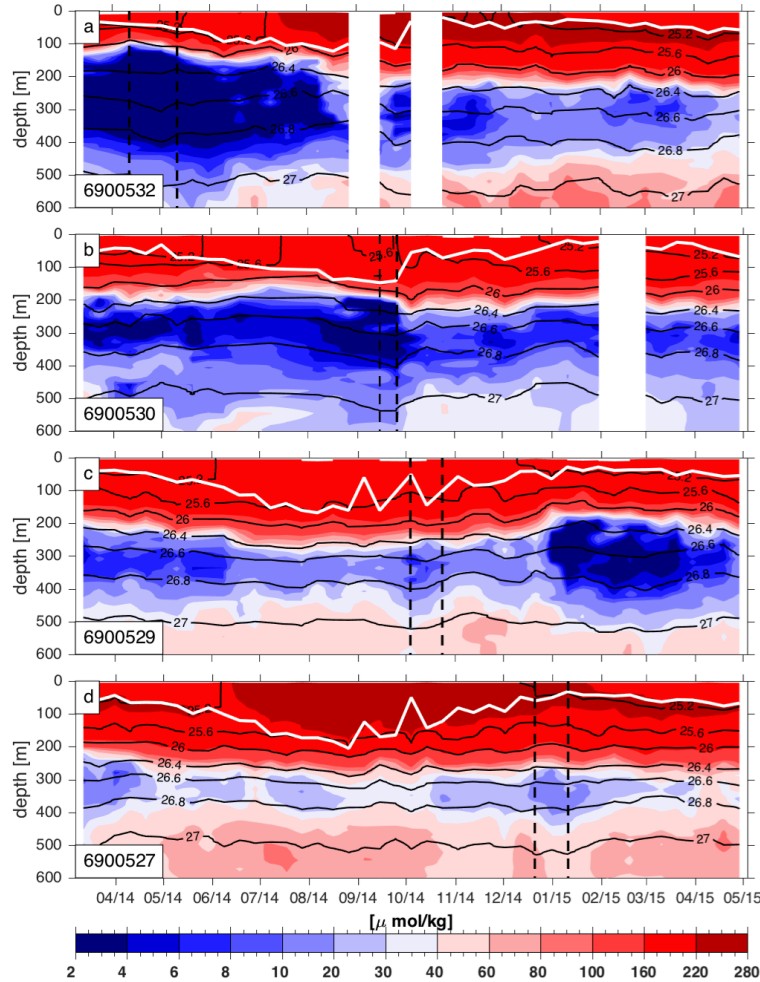

**Figure 7.** Distribution of oxygen (in μmol kg$^{-1}$, coloured) and density (black lines) vs time in the upper 600 m depth of floats 6900532 (a), 6900530 (b), 6900529 (c) and 6900527 (d) which have been trapped within the ACE2 at different stages. The residence time of the floats in April/May 2014, September 2014, October 2014 and December 2014/January2015 is marked by dashed black lines. The white curve is the mixed layer depth defined for the depth where the potential density anomaly is 0.125 kg m$^{-3}$ larger than at the surface.





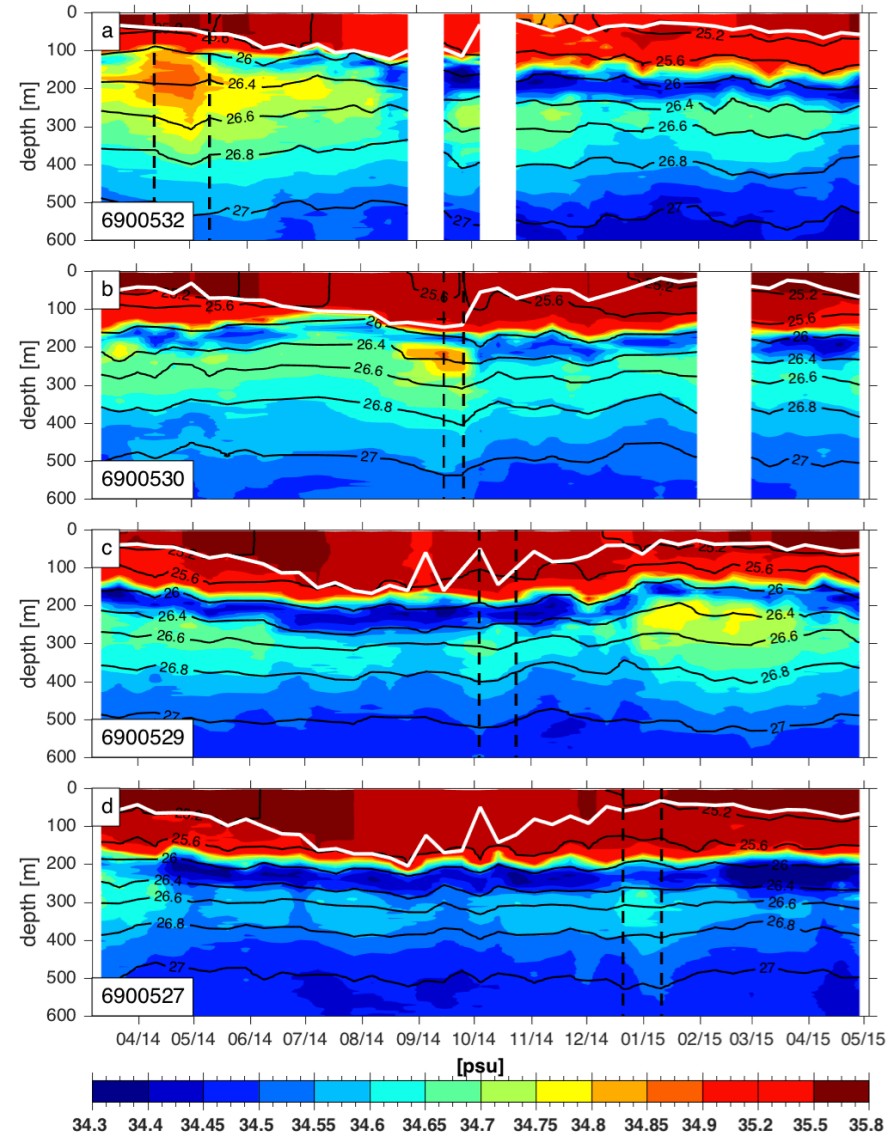

**Figure 8.** Same as Fig. 7 but for salinity distribution.



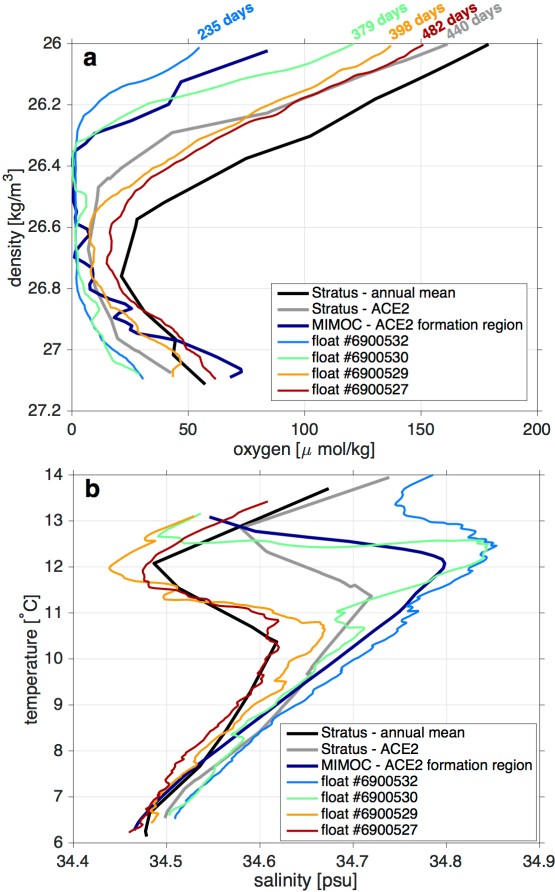

**Figure 9.** Profiles of a) oxygen vs density and b) TS-diagrams at the Stratus mooring (84°57'W, 19°37'S) for a 1-year-mean (10 April 2014 to 9 April 2015; black) and during the passage of the ACE2 in November 2014 (grey line), from the estimated formation region at 71°W, 23°S from the MIMOC

5  climatology (dark blue) and from data of four floats (6900527, -29, -30 and -32) trapped in the ACE2 (for coloured lines see legend). The age of the ACE2 in days during the respective measurement is indicated in the upper panel. The age of the ACE2 [in days] at the time of the respective float measurements is marked in the upper panel.