# Peer review of "Transport, properties and life-cycles of mesoscale eddies in the eastern tropical South Pacific"

_Ocean Science, 2018_

## Referee Comment (RC1) · Anonymous Referee #1 · 5 Feb 2018

Review of Manuscript: " Impact of mesoscale eddies on water mass and oxygen distribution in theeastern tropical South Pacific" byRena Czeschel et al.

This paper used multiple-platform observations to investigate different types of mesoscale eddies in the eastern tropical South Pacific region. The transports and life-cycle evolution of the observed eddies is the main point of this paper, which provide with a lot of valuable information of the eddies within this region. The results of this paper is abundant and interesting, if the authors can put the observational results into a more consistent story, the significance of this paper will be improved substantially. Some further clarification and discussion are also needed. I recommend considering

publication after major revision, andI will leave my questions in the Specific Comments.

Specific Comments:

1. Page2 Line3-5: It is a little confusing to say the "isolation" and "mixing" of the water mass at the same time. If the mixing reaches maximum, the water mass is no longer strictly isolated. Maybe the reader can understand the meaning of authors after finishing the whole paper, as the presentation here within the abstract, the author could give a clarified presentation. 2. Page5 Line 14-17: A sketch of the mooring and the equipment/probes on it will help readers to imagine how the mooring operates. 3. Page7 Line 5-9: The figure of the trajectory of the Argo floats should be given, which will directly demonstrate how the floats being trapped and moving with the eddy. 4. Page8 Line 1-6: The high-frequency observation of mesoscale eddies by mooring is quite valuable. With the altimeter data providing the eddies' location and radius, the mooring observations can be projected onto eddy center coordinate and reconstruct the three-dimensional structure of the mesoscale eddies (Similar to Fig.4), which may give a lot of useful information of the eddies. 5. Page12 Line14-17: There is two types of transports can be done by eddies: stirring and trapping. The stirring transport happen when there is a background tracer gradient, with the swirling velocity of eddies, the down-gradient transport of tracer will emerge. The stirring transport does NOT need the eddy to move. On the other hand, when eddy traps a water mass within its core area, the tracer within this water mass will be transported. The net flux of this kind of trapped-transport happens when the eddy is moving and the trapped water mass having different properties contrasted with surrounding environment. The main focus of the authors is the second kind of transportation. This should be clarified. And with the measurements of the mooring, the first kind of stirring transport can also be evaluated quantitatively. 6. Page18 Line 1-5: From Fig.5c, significant variation of the nonlinear parameter U/c can be observed. At the same time, the nonlinear parameter U/c is also used by the authors to compute the vertical extent of the trapped fluid by the eddy. This means the volume of the trapped water by the eddy will also

experience significant variation. But the trapped water mass is expected to be quite coherent and isolated, what will cause significant variation of its volume. The authors should give further clarification and discussion. 7. Page 22 Line6-8: The lateral mixing between eddy-core and surrounding water is related to the evolution of eddy, especially its decaying processes. Could the lateral mixing derived from the Argo floats be used to explain the variation of eddy amplitude observed by altimeter?

————————————————

---

## Referee Comment (RC2) · Anonymous Referee #2 · 16 Mar 2018

The manuscript explores the consequences of eddies transporting properties out of the low oxygen upwelling region associated with the Peruvian-Chilean coast and into the oligotrophic open S.Pacific. Transport by coherent eddies rather than by mean flow (or lateral mixing associated with turbulent flow around eddies) is potentially an important component of any attempt to budget properties within a region. Hence it is of significant interest in understanding the controls on a low oxygen region such as the one studied. While the authors bring a lot of detail to the study I feel that they lose track a little of this broader intention and don't really match up to their title as well as they could at present. My main suggestions are hopefully useful ones to bridge this gap.

First, the fluxes associated with the eddies need context. The area of interest needs to be more clearly defined at the start and the reasons given for these choices. At present some of this key information is not presented until p15. The authors should then provide westward fluxes out of the region due to the mean flow. The westward mean current can be calculated from the mooring data. 'Outside' eddy profiles can be used for mean tracer properties (salt, temperature, oxygen) if there are concerns over simply averaging them introducing biases.

Second, the flux estimates need to include uncertainties. I'm not suggesting that the authors are in a position to accurately estimate the eddy transport - they themselves acknowledge that it is not possible given the small sample size. However, the numbers they give still need to come with likely uncertainties to be of wider use. Two several sources of uncertainty in particular come to mind:

-Definition of the eddy core - The mean diameter between filtered max north and south velocities is used so the standard deviation could provides estimates of uncertainty. I would also question whether the maximum velocities are good indicators of the eddy boundary. The position of maximum velocity is an area of considerable angular shear and hence potentially of lateral mixing. The extent of solid body rotation (i.e. the radius to which velocity still increases linearly) could be argued to be a better definition for the purposes of coherent eddy transport.

- Observations taken offset from the eddy centre - using an eddy thought to be sampled close to the centre would allow this to be assessed.

Third, judging from the movie there is not just variability in eddy properties passing through Stratus. There may be variability in the number and intensity of eddies crossing different parts of the north-south line through Stratus. This affects how well Stratus can be viewed as a position to monitor these eddy fluxes. The satellite data could be used to contruct plots of numbers and mean anomalies of eddies crossing in a period versus latitude along this line. Ideally the plot would be flat give or take the inevitable noise but
big peaks would suggest potential bias. For the existing analysis in the manuscript the authors need to show how sensitive their analysis of the satellite data is to the choice of 7 days visibility and 45-150km (p9, line 8)

I also have a number of more specific points:

- the paper would be benefit from a schematic showing the regions and boundaries of interest plus mean (rather than eddy) fluxes

- the properties of the OMZ need to be stated early on - typical depth range, oxygen concentration, horizontal extent

- there is considerable blurring of results and discussion. For example p9, l21 to p10, l11 would be better in a discussion. Also, p14, lines 3 and 9 and other places.

- the authors rightly point out the difficulty in separating seasonal from other variability given the existing data but it would be good to see some discussion of what would be necessary to allow these to be separated in the future.

- the discussion of what has been lost on p16, lines 1-9 is a little confusing. Given the choice of the Stratus mooring effectively as a monitoring station on the boundary of the region it seems odd to discuss what has bene lost when when the eddies have reached this point rather than what they are carrying beyond it. What they've already lost is irrelevant to the budget given the focus on Stratus.

Minor stuff:

- p3, line 7: "in different"
- p6, lines 12-14: cut "has taken over...scientific content" as not relevant to paper
- p7, line 6: cut 'were'
- p8, lines 8-9: A fuller explanation of how this was done is needed
- p8, line 10: cut "of"

OSD
- p10, line 22: "typical" by what metric?

- p10, lines 24 and 25 and Fig 3: this isn't very clear in the figure.

- p11, lines 5 and 20: why was 183m chosen?
- p16, lines 12-14: need a fuller explanation

- p17, lines 17-18: Chlorophyll concentration does not imply growth. Units of production are not mg/m-3. The phytoplankton community could have grown or been entrained as far back as the formation region/time.

- p18, lines 5-11: ability to estimate Ro accurately is very dependent on the resolution of sampling. I would be more cautious unless you can show that your sampling resolution would not have biased the result.

- p19, line 4: reference needed for this statement on lateral mixing

- p19, lines 6-7. High velocity in itself is not a guarantee. The circular flow of an eddy needs to be taken into account as considerable rotational/angular shear will take place in high velocity areas outside the region of solid body rotation.

p20, lines 12-18: this is not very clear e.g. "too high for heat" relative to what?

p20, line 21: AHA twice

p21, lines 5 to 8: if the eddies are exporting oxygen deficit from the OMZ region aren't they eroding rather than maintaining it?

p21, line 14: highest amount globally? Based on what bearing in mind that proper diagnosis requires vertical profiles?

p21, lines 22-23: Has this feature of the model been verified in any way?

p22, lines 11-12: need explanation of what sort of mixing and a reference.

p22, line 26: reference needed for the mean circulation values
p23, line 5: " main contributor" is a strong claim. By what metric? Supplementary material, fig S2: the caption needs to be clearer.

---

## Author Comment (AC1) · 9 May 2018

Anonymous Referee #1 Received and published: 5 February2018 Review of Manuscript: "Impact of mesoscale eddies on water mass and oxygen distribution in the eastern tropical South Pacific" by Rena Czeschel et al.

This paper used multiple-platform observations to investigate different types of

mesoscale eddies in the eastern tropical South Pacific region. The transports and life-cycle evolution of the observed eddies is the main point of this paper, which provide with a lot of valuable information of the eddies within this region. The results of this paper is abundant and interesting, if the authors can put the observational results into a more consistent story, the significance of this paper will be improved substantially. Some further clarification and discussion are also needed. I recommend considering publication after major revision, and I will leave my questions in the Specific Comments.

Reply to reviewer #1 We would like to thank the reviewer for taking the time and for providing constructive and very specific comments, which helped to improve the manuscript considerably. The revised manuscript emphasizes the transport of eddies and the title has been changed to "Transport, properties and life-cycles of mesoscale eddies in the eastern tropical South Pacific". The "Discussion and conclusion" paragraph has been changed to "Discussion and outlook" and restructured. We have carefully addressed his/her comments. The point-by-point responses follow below (written in bold).

Note: During the review process we noted the failure of the temperature sensors in 70, 78 and 280 m in the Stratus mooring in November 2014, September 2014 and March 2015, respectively. For calculations of the annual mean as a background field only data were used that covered an entire year (10 April 2014 to 9 April 2015). Therefore we skipped these temperature data as well as the uncomplete salinity data in 85 m depth for the calculation of the annual mean, which lead to slight modifications. Therefore Fig. 6a,b and Fig. 9b has been refigured and calculations of the AHA and ASA (Table 1), the heat and salt transport across the Stratus mooring (Table 2) as well as heat and salt fluxes in the ETSP (chapter 4.3) have been corrected. We corrected the eddy track of ACE2 for the first weeks, which is shown in the movie (supplement) and Fig. 1. Consequently, the composite of the ACE2 of the surface signatures for SLA, SST anomaly and chlorophyll have slightly changed (Fig. 4c, g, k) as well as the properties of ACE2 in Fig. 5.
Specific Comments:

1. Page 2, Line 3-5: It is a little confusing to say the "isolation" and "mixing" of the water mass at the same time. If the mixing reaches maximum, the water mass is no longer strictly isolated. Maybe the reader can understand the meaning of authors comment after finishing the whole paper, as the presentation here within the abstract, the author could give a clarified presentation.

We rewrote the sentence to: "Furthermore, four profiling floats were trapped in the ACE2 during its westward propagation between the formation region and the open ocean, which allows conclusions on lateral mixing of water mass properties with time between the core of the eddy and the surrounding water. Strongest lateral mixing was found between the seasonal thermocline and the eddy core during the first half of the eddy lifetime."

2. Page 5, Line 14-17: A sketch of the mooring and the equipment/probes on it will help readers to imagine how the mooring operates.

We added a sketch of the mooring and the distribution of all instruments in the supplementary Fig. S1. Consequently we changed the consecutive numbering of the corresponding figures S1-S3 to S2-S4 in the figure caption as well as in the text.

Fig. S1. Distribution of the instruments attached to the Stratus mooring between 8 March 2014 to 25 April 2015.

3. Page 7, Line 5-9: The figure of the trajectory of the Argo floats should be given, which will directly demonstrate how the floats being trapped and moving with the eddy.

We have followed your suggestions. In order to improve the visualization we added the track of eddy ACE2 as well as the trajectories of the relevant floats into the movie (supplementary) to better follow the movements of the floats within the eddy instead of a new figure (showing the trajectories of the floats that are trapped in the eddy). The locations of the floats that cross the eddies are also marked in the composite of the

eddies (Fig. 4). We hope that you agree with this approach.

4. Page 8, Line 1-6: The high-frequency observation of mesoscale eddies by mooring is quite valuable. With the altimeter data providing the eddies' location and radius, the mooring observations can be projected onto eddy center coordinate and reconstruct the three-dimensional structure of the mesoscale eddies (Similar to Fig.4), which may give a lot of useful information of the eddies.

We have tried to follow your idea of a three-dimensional plot. The figures below show the time series of the temperature anomalies vs depth at the position of the Stratus mooring (19°37'S, 84°57'W) and the Hovmoeller diagram (time-latitude) of sea surface temperature anomaly (coloured) and sea level anomaly (contoured) at the longitude position of the Stratus mooring for the a) MWE, b) ACE1, c) ACE2, and d) CE. We do not think that these plots would enhance the amount of information about the eddies significantly. Instead, we added a Hovmoeller-diagramm (time-latitude) of the SLA at the longitude position of the Stratus mooring (Fig. 3b, see below) showing the position of the individual eddies in relation to the mooring (white dashed line).

Figure 3. Time series for the deployment period 8 March 2014 to 25 April 2015 at the position of the Stratus mooring (19°37'S, 84°57'W) for (a) weekly delayed, high-pass filtered sea level anomaly (in cm; blue curve) and geopotential anomaly between 450 and 295 m depth in m2 s-2 (orange curve), (c) oxygen in $\mu$mol kg-1, (d) salinity and (e) the meridional velocity component in cm s-1, Hovmoeller diagram (time-latitude) at the longitude position of the Stratus mooring for (b) SLA in cm. The white curve in (e) is the mixed layer depth defined for the depth where the potential density anomaly is 0.125 kg m-3 larger than at the surface. The black dots on the vertical line at the left mark the depths of the used oxygen (c), conductivity (d) and velocity (e) sensors and the black contour lines are selected density contours. Black solid (dashed) lines show the date of the passages of the anticyclonic (cyclonic) eddies. 5. Page 12, Line 14-17: There is two types of transports can be done by eddies: stirring and trapping. The stirring transport happen when there is a background tracer gradient, with the swirling velocity

of eddies, the down-gradient transport of tracer will emerge. The stirring transport does NOT need the eddy to move. On the other hand, when eddy traps a water mass within its core area, the tracer within this water mass will be transported. The net flux of this kind of trapped-transport happens when the eddy is moving and the trapped water mass having different properties contrasted with surrounding environment. The main focus of the authors is the second kind of transportation. This should be clarified. And with the measurements of the mooring, the first kind of stirring transport can also be evaluated quantitatively.

Thank you very much, you are absolutely right. We have added a few sentences in the manuscript to clarify that: "Horizontal eddy transport can be explained by two mechanisms: 1) by eddy stirring, which occurs at the periphery of the eddy (e.g. Gaube et al., 2015; Chelton et al., 2011) and 2) by eddy transport of water masses trapped in the eddy interior (Gaube 2013; 2015). We are focusing on the latter mechanism. "

6. Page 18, Line 1-5: From Fig.5c, significant variation of the nonlinear parameter U/c can be observed. At the same time, the nonlinear parameter U/c is also used by the authors to compute the vertical extent of the trapped fluid by the eddy. This means the volume of the trapped water by the eddy will also experience significant variation. But the trapped water mass is expected to be quite coherent and isolated, what will cause significant variation of its volume. The authors should give further clarification and discussion.

Yes, you are right and now we have discussed this in the text on p18: "Nonetheless, significant variations of the nonlinear parameter U/c determined at the surface might indicate changes of the volume of the eddies, which can be influenced by friction, stratification, fluctuations of the mean flow or the collapse with other eddies. Maps of SLA show a permanent change of the radius due to an irregular and varying shape and the merging with other eddies (supplement: Movie M1), which makes it sometimes difficult to track an eddy during its whole lifetime. Fluctuations are also produced by the coarse resolution of the satellite data ($\frac{1}{4}°$ x $\frac{1}{4}°$) and the merging algorithms used by AVISO.

However, the MWE and CE show stronger fluctuations of the nonlinear parameter than the ACE1/2, which probably mirrors the higher variability of the swirl velocity of both eddy types (Fig. 5b). Nonetheless, the nonlinear parameter U/c is always higher than 1 and therefore indicates a trapped volume despite strong fluctuations at the surface. The small variations of the eddy properties of the ACE2 (Fig. 5a-e) suggest a relatively stable structure."

7. Page 22, Line 6-8: The lateral mixing between eddy-core and surrounding water is related to the evolution of eddy, especially its decaying processes. Could the lateral mixing derived from the Argo floats be used to explain the variation of eddy amplitude observed by altimeter?

Thanks for the hint. For a better overview we marked the residence time of the four floats that were trapped in ACE2 in Fig. 5a and discussed the mixing with the observations from altimeter data (p23). "During this period the variability of the amplitude of the ACE2 is negligibly small, which might be due to the coarse resolution of the satellite data. The radius increases up to 50 km with a nearly consistent rotation velocity at the same time (Fig. 5a, b). During the second half of the lifetime the radius of the ACE2 slightly increases but the maximal rotation velocity and therefore the nonlinear parameter decreases until the decay of the ACE2."

Please also note the supplement to this comment:
https://www.ocean-sci-discuss.net/os-2018-5/os-2018-5-AC1-supplement.zip

[Figure]

[Figure]

**Fig. 1.**

[Figure]

Fig. 2.

[Figure]

**Fig. 3.**

[Figure]

Fig. 4.

[Figure]

**Fig. 5.**

[Figure]

Fig. 6.

**Fig. 7.**

**Fig. 8.**

[Figure]

**Fig. 9.**

---

## Author Comment (AC2) · 9 May 2018

Anonymous Referee #2 Received and published: 16 February2018

The manuscript explores the consequences of eddies transporting properties out of the low oxygen upwelling region associated with the Peruvian-Chilean coast and into the oligotrophic open S. Pacific. Transport by coherent eddies rather than by mean

flow (or lateral mixing associated with turbulent flow around eddies) is potentially an important component of any attempt to budget properties within a region. Hence it is of significant interest in understanding the controls on a low oxygen region such as the one studied. While the authors bring a lot of detail to the study I feel that they lose track a little of this broader intention and don't really match up to their title as well as they could at present. My main suggestions are hopefully useful ones to bridge this gap.

Reply to reviewer #2 We would like to thank the reviewer for his/her helpful comments, which helped to improve the manuscript during the revision. The revised manuscript emphasizes the transport of eddies and the title has been changed to "Transport, properties and life-cycles of mesoscale eddies in the eastern tropical South Pacific". The "Discussion and conclusion" paragraph has been changed to "Discussion and outlook" and restructured. We have modified the manuscript to address his/her comments. A detailed response follows below (written in bold).

Note: During the review process we noted the failure of the temperature sensors in 70, 78 and 280 m in the Stratus mooring in November 2014, September 2014 and March 2015, respectively. For calculations of the annual mean as a background field only data were used that covered an entire year (10 April 2014 to 9 April 2015). Therefore we skipped these temperature data as well as the uncomplete salinity data in 85 m depth for the calculation of the annual mean, which lead to slight modifications. Therefore Fig. 6a,b and Fig. 9b has been refigured and calculations of the AHA and ASA (Table 1), the heat and salt transport across the Stratus mooring (Table 2) as well as heat and salt fluxes in the ETSP (chapter 4.3) have been corrected. We corrected the eddy track of ACE2 for the first weeks, which is shown in the movie (supplement) and Fig. 1. Consequently, the composite of the ACE2 of the surface signatures for SLA, SST anomaly and chlorophyll have slightly changed (Fig. 4c, g, k) as well as the properties of ACE2 in Fig. 5.

First, the fluxes associated with the eddies need context. The area of interest needs to be more clearly defined at the start and the reasons given for these choices. At

present some of this key information is not presented until p15. The authors should then provide westward fluxes out of the region due to the mean flow. The westward mean current can be calculated from the mooring data. 'Outside' eddy profiles can be used for mean tracer properties (salt, temperature, oxygen) if there are concerns over simply averaging them introducing biases.

Results of the eddy heat fluxes have been placed in a wider context (see chapter 4.3) and the investigation of fluxes associated with eddies is now mentioned in the introduction. "Knowledge about the initial eddy-core conditions near the generation areas, measurements during the mid-age of the eddy due to Argo floats and measurements of the Stratus mooring at the end of the eddy lifetime allows us to investigate the fluxes associated with the eddies and the lateral mixing from the eddy-core water masses with its surrounding waters. We think that the area of interest is well described and reasons for this choice are described in detail in the Introduction ("Peruvian upwelling region", "largest eddy frequency in the ETSP", "Stratus ORS mooring is located in the transition zone between the OMZ and the well-oxygenated subtropical gyre", "weak mean currents" and many more besides). Westward fluxes are shown in the supplement (Fig. S4, annual and multi-year mean of zonal and meridional velocity at the Stratus mooring) and are discussed in the chapter 5 "Discussion and outlook". The one-year mean (10 April 2014 to 9 April 2015) of tracer properties (salinity, temperature and oxygen) at the Stratus mooring is already used as 'outside' eddy profile for the calculation of the anomalies trapped within the eddies (Fig. 6). The method has been described in chaper 3.1 and the annual mean of oxygen and a T-S diagram at the Stratus mooring is shown in Fig. 9.

Second, the flux estimates need to include uncertainties. I'm not suggesting that the authors are in a position to accurately estimate the eddy transport - they themselves acknowledge that it is not possible given the small sample size. However, the numbers they give still need to come with likely uncertainties to be of wider use. Two several sources of uncertainty in particular come to mind:
- Definition of the eddy core -The mean diameter between filtered max north and south velocities is used so the standard deviation could provides estimates of uncertainty. I would also question whether the maximum velocities are good indicators of the eddy boundary. The position of maximum velocity is an area of considerable angular shear and hence potentially of lateral mixing. The extent of solid body rotation (i.e. the radius to which velocity still increases linearly) could be argued to be a better definition for the purposes of coherent eddy transport.

Interesting point and yes, it is worth discussing different indicators to estimate the size of an eddy. Taking the maximum velocity for the eddy boundary is a common indicator in comparative studies (Chaigneau et al. 2011). For the comparison of the calculated AHA, ASA, AOA with studies from Chaigneau et al. (2011) it is necessary to use the same method. We write in the text (p8): "Often the eddy boundary is defined as the streamline with the strongest swirl velocity (for more information on such an eddy detection algorithm see e.g. Nencioli et al., 2010). For comparison of our results with the results of e.g. Chaigneau et al. (2011) we also use the boundary definition of the streamline with the strongest swirl velocity." Additionally, we calculated error bars for the radius, volume and transport of the eddies (see table 1 and text, p8): "Error bars for the horizontal eddy boundaries are computed using the mean of the maximum absolute values of the hourly-mean southward and northward velocity. As a result the swirl velocity increases and likewise the vertical extent of the eddies due to the ratio between swirl velocity U and propagation velocity c. Nonetheless, the deviations of the horizontal boundaries of the eddy are small. The deviations of the radius are used to estimate the error for AHA, ASA and AOA from uncertainties of the size of the eddies."

- Observations taken offset from the eddy centre -using an eddy thought to be sampled close to the centre would allow this to be assessed. Generally, this is a good idea. Due to the strong development of the eddies during their westward propagation the offset would be probably time dependent. Additionally, due to the coarse temporal and spatial resolution the centre of the eddy cannot be located precisely. Therefore, at present our

observations do not allow us to assess an offset.

Third, judging from the movie there is not just variability in eddy properties passing through Stratus. There may be variability in the number and intensity of eddies crossing different parts of the north-south line through Stratus. This affects how well Stratus can be viewed as a position to monitor these eddy fluxes. The satellite data could be used to contruct plots of numbers and mean anomalies of eddies crossing in a period versus latitude along this line. Ideally the plot would be flat give or take the inevitable noise but big peaks would suggest potential bias. For the existing analysis in the manuscript the authors need to show how sensitive their analysis of the satellite data is to the choice of 7 days visibility and 45-150km (p9, line 8)

This is a very nice idea for an extra paper or for a study with model data. Determining the seasonal or interannual variability of heat fluxes from eddies would be beyond the scope of our manuscript. We are discussing the seasonal variability of the eddies generated off the coast (Fig. 2) and the eddy coverage in the ETSP (Fig. 1). We added estimations about the number of eddies dissipating within the transition zone which is located around the Stratus mooring.

I also have a number of more specific points: - the paper would be benefit from a schematic showing he regions and boundaries of interest plus mean (rather than eddy) fluxes We calculated the eddy fluxes for anticyclones and cyclones more specifically for an offshore area in the ETSP and heat transport estimates of Chaigneau et al. (2008) into context with the results. Changes in the text on page 15: "Available anomalies of heat, salt, and oxygen of cyclonic and anticyclonic eddies gained from the Stratus mooring and from the literature (Table 1) are now used to estimate the relative contribution of long-lived eddies to fluxes of mass, heat, salt, and oxygen in an offshore area of the ETSP. The mean heat (in W), salt (in kg s-1) and oxygen transport ($\mu$mol s-1) are calculated by multiplying the amount of AHA, ASA, and AOA of the composite eddies with the number of eddies dissipating per year in an offshore area (corresponding to a flux divergence). We define an area reaching in north-south direction from 10°-24°S.

The transition area is bordered in the east by a line running parallel to the Peruvian and Chilean coast at a distance of 6° and in the west by the 90°W-longitude corresponding to a size of ~1.7 x 106 km2. Based on averaged satellite measurements 58.6 eddies of all eddies that are generated off the coast (Fig. 2) reach the offshore area per year from which 28.9 are cyclones and 29.7 are anticyclones. 2.1 cyclones, and 0.7 anti-cyclones and mode-water eddies propagate into the area west of the 90°W longitude meaning that 26.8 of the cyclones and 29 of the anticyclones and mode-water eddies have dissipated and therefore transported a certain amount of heat, salt, and oxygen into the offshore zone. Based on the mean of AHA, ASA, and AOA for the composite eddies the mean transport of heat (salt, oxygen) per year from the coastal region into the transition zone is -6.4 x 1012 W (-2.4 x 105 kg s-1, -5.7 x 1010 $\mu$mol kg-1 s-1) for cyclones and 4.7 x 1012 W (1.5 x 105 kg s-1, -5.9 x 1010 $\mu$mol kg-1 s-1) for anticy-clones and mode-water eddies in agreement with estimates for transport anomalies of heat and salt in this region by Chaigneau et al., 2011. "

- the properties of the OMZ need to be stated early on -typical depth range, oxygen concentration, horizontal extent We rewrote relevant parts of the Introduction (p4): "In general, the large-scale oxygen distribution in the ETSP is dominated by a strong OMZ at depths of 100-900 m with minimum oxygen values at about 350 m depth ($\sigma\theta$=26.8 kg m-3) and suboxic conditions of <4.5 $\mu$mol kg-1 off Peru (e.g., Karstensen et al., 2008; Paulmier and Ruiz-Pino, 2009 or Fig. 1b). In the OMZ the oxycline of 60 $\mu$mol kg-1 extends along the South American coast from 35°S to the equator where it reaches westward to nearly 160°E (Llanillo et al. 2018). "

- there is considerable blurring of results and discussion. For example p9, l21 to p10, l11 would be better in a discussion. Also, p14, lines 3 and 9 and other places. We modified the text of the manuscript and we hope that it is now easier to follow. E.g., we moved part of the text on p9 l21 to p10, l11 (first text version) to the Discussion and removed part of the text as it was redundant with text in the Discussion. Also part of the former p14, lines 3 to 9 was moved to the Discussion.

- the authors rightly point out the difficulty in separating seasonal from other variability given the existing data but it would be good to see some discussion of what would be necessary to allow these to be separated in the future. The "Discussion and conclusion" paragraph was changed to "Discussion and outlook" and an outlook included what would be necessary for separating the variability components: "The most promising methods to tackle the open questions would be long-term measurements near the coast, glider measurements in the center of the eddies and accompanying model investigations".

- the discussion of what has been lost on p16, lines 1-9 is a little confusing. Given the choice of the Stratus mooring effectively as a monitoring station on the boundary of the region it seems odd to discuss what has bene lost when when the eddies have reached this point rather than what they are carrying beyond it. What they've already lost is irrelevant to the budget given the focus on Stratus. The text on page 16 was modified and hopefully is no longer confusing. "Both types of eddies show negative oxygen fluxes in the layer defined as the depth of the coherent structure of the eddy (Table 1) meaning that anticyclones and cyclones transport less oxygenated water into the upper and middepth open ocean and therefore have an impact on the balance and size of the OMZ in the ETSP, which is also confirmed by models (Frenger et al., 2018)."

Minor stuff:

-p3, line 7: "in different" Done, thanks.

-p6, lines 12-14: cut "has taken over...scientific content" as not relevant to paper Done.

-p7, line 6: cut 'were' Done

-p8, lines 8-9:A fuller explanation of how this was done is needed The sentence has been rewritten: "Assuming a symmetric eddy, the centre of the MWE passed the mooring on 8 March 2014 and fully passed the mooring until end of March 2014. The measurements of the eastern part of the eddy during that time span were mirrored to

obtain the full coverage of the MWE."

-p8, line 10: cut "of" Done

-p10, line 22: "typical" by what metric? The sentence has been rewritten: "The mooring instruments recorded the parameter distribution at the southern rim of the MWE revealing anomalous low oxygen of less than 10 $\mu$mol kg-1 and anomalous high salinity (temperature) of more than 34.65 psu (10.6°C) in the eddy core in 300 m depth. It was accompanied with an upward bending of isopycnals above ($\sim$250 m depth) and downward bending beneath ($\sim$350 m depth) the eddy core (Fig. 3c, d), which is typical for a mode water eddy in contrast to anticyclonic and cyclonic eddies."

-p10, lines 24 and 25 and Fig 3: this isn't very clear in the figure. You are right, thanks. The text has been modified: "The mooring instruments recorded the parameter distribution at the southern rim of the MWE revealing anomalous low oxygen of less than 10 $\mu$mol kg-1 and anomalous high salinity (temperature) of more than 34.65 psu (10.6°C) in the eddy core in 300 m depth. It was accompanied with an upward bending of isopycnals above ($\sim$250 m depth) and downward bending beneath ($\sim$350 m depth) the eddy core (Fig. 3c, d), which is typical for a mode water eddy in contrast to anticyclonic and cyclonic eddies."

-p11, lines 5 and 20: why was 183 m chosen? We wanted to show the impact of eddies on water mass properties in the near surface layer, which is not influenced by variability in the mixed layer. As in the region of the Stratus mooring the mixed layer reaches down to about 160 m depth (Czeschel et al. 2015) the instrument below the mixed layer in 183 m depth was chosen.

-p16, lines 12-14: need a fuller explanation The negative oxygen fluxes by cyclones and anticyclones are defined for and dependant of the layer defined as the depth of the coherent structure of the eddy (Table 1), now mentioned in the text. The model investigation mentioned is published and the reference changed to Frenger et al. 2018 instead of pers. com. "Both types of eddies show negative oxygen fluxes in the layer
defined as the depth of the coherent structure of the eddy (Table 1) meaning that anticyclones and cyclones transport less oxygenated water into the upper and middepth open ocean and therefore have an impact on the maintenance and size of the OMZ in the ETSP, which is also confirmed by models (Frenger etal., 2018 ).”

-p17, lines 17-18: Chlorophyll concentration does not imply growth. Units of production are not mg/m-3. The phytoplankton community could have grown or been entrained as far back as the formation region/time. Thank you for this information, it should have been chlorophyll concentration instead of chlorophyll production, this is now corrected.

-p18, lines 5-11: ability to estimate Ro accurately is very dependent on the resolution of sampling. I would be more cautious unless you can show that your sampling resolution would not have biased the result. You are right, regarding the resolution of 0.25° x 0.25° of the underlying SLA data the statements are too strong. We rewrote the mentioned paragraph as followed: “This is consistent with our observations showing constant Rossby numbers of less than 0.1 between 0.2 and 0.8 reflecting a stable, geostrophic phase phase over 60 % of the lifetime of all eddy types. At the beginning and at the end of the eddy lifetime the Rossby numbers are increasing and indicating the influence of possible ageostrophic processes. But the increase of the Rossby number for both anticyclones is not as striking as for the MWE and CE. However, a very detailed and exact discussion about the evolution of the Rossby Radius (and also the radius, the average maximal rotation and the nonlinearity) and possible ageostrophic processes is not possible due to the coarse resolution of the underlying SLA data.”

-p19, line 4: reference needed for this statement on lateral mixing The sentence has been modified.

-p19, lines 6-7. High velocity in itself is not a guarantee. The circular flow of an eddy needs to be taken into account as considerable rotational/angular shear will take place in high velocity areas outside the region of solid body rotation. You are right. The sentence has been reworded: “This density level corresponds to a depth between 100

and 170 m, where a high swirl velocity exists within the ACE2 (Fig. 6a), which is essential to keep up the coherent structure."

p20, lines 12-18: this is not very clear e.g. "too high for heat" relative to what? The passage has been clarified and modified due to the corrected number for the AHA: "In this study the negative respectively positive heat anomalies of the CE (-8.9 x1018 J), ACE2 (8.1 x1018 J) and MWE (1.0 x1018 J) almost balance each other. In contrast, the sum of negative respectively positive salt anomalies transported within the CE (-41.5 x1010 kg), ACE2 (25.2 x1010 kg), and MWE (-3.1 x1010 kg) is unbalanced."

p20, line 21: AHA twice Done, thanks.

p21, lines 5 to 8: if the eddies are exporting oxygen deficit from the OMZ region aren't they eroding rather than maintaining it? Now we say "balancing" instead of "maintaining".

p21, line 14: highest amount globally? Based on what bearing in mind that proper diagnosis requires vertical profiles? The observations by Zhang et al. 2017 are based on Argo floats and the highest amount of MWE is shown in their Figure 2, now mentioned in the revised text. "According to a global investigation of Argo floats the eastern South Pacific off Peru and Chile seems to have the highest amount of MWEs, which are also deep reaching compared to other regions (Zhang et al., 2017; their Fig. 2)."

p21, lines 22-23:Has this feature of the model been verified in any way? The results from Pegliasco are from observations. We included a reference for a observation/model comparison by Kurian et al., 2011. "Observations and model results for the Calfornia Current system showed a good agreement between observed and modelled eddy structures (Kurian et al. 2011)."

p22, lines 11-12: need explanation of what sort of mixing and a reference. The statement has been modified and we have added a reference: "Stronger mixing during the first half of the eddy lifetime could be related to a stronger wind curls near the coast in

comparison to the open ocean (Albert et al. 2010, their Fig. 1b)."

p22, line 26: reference needed for the mean circulation values Reference Ayon et al. 2008 included for mean circulation.

p23, line 5: " main contributor" is a strong claim. By what metric? The sentence has been reworded: "Observations from the mooring deployment period March 2014 to April 2015 and profiling floats deployed near eddies in March 2014 (Fig. 1) show the importance of eddies in the weak flow region of the ETSP as mesoscale eddies play a crucial role in water mass distribution."

Supplementary material, fig S2: the caption needs to be clearer. Figure caption has been rewritten. "Figure S3. Vertical profiles from data of four floats during their stay within the ACE2 at different stages of its lifetime (6900532: day 235, 6900530: day 379, 6900529: day 398, and 6900527: day 482). Shown are the changes per day of a) oxygen, b) temperature, and c) salinity on density surfaces between float 6900530 and 6900532 (blue), between float 6900529 and 6900530 (green) and between float 6900527 and 6900529 (red)."

Please also note the supplement to this comment:
https://www.ocean-sci-discuss.net/os-2018-5/os-2018-5-AC2-supplement.zip

[Figure]

**Fig. 1.**

[Figure]

**Fig. 2.**

[Figure]

**Fig. 3.**

[Figure]

**Fig. 4.**

a) **radius = r**

b) **average maximal rotation = u**

c) **nonlinearity Parameter**

d) **Rossby Number**

e) **centered SLA**

f) **centered SST**

**normalized eddy lifetime**

**Fig. 5.**

[Figure]

OSD

Interactive
comment

[Figure]

[Figure]

**Fig. 6.**

**Fig. 7.**

**Fig. 8.**

[Figure]

**Fig. 9.**